# Control of bacterial cell wall autolysins by peptidoglycan crosslinking mode

Laura Alvarez [1,10], Sara B. Hernandez[1,8,10], Gabriel Torrens [1], Anna I. Weaver [2,3,9], Tobias Dörr [2,3,4] & Felipe Cava [1,5,6,7] ✉

To withstand their internal turgor pressure and external threats, most bacteria have a protective peptidoglycan (PG) cell wall. The growth of this PG polymer relies on autolysins, enzymes that create space within the structure. Despite extensive research, the regulatory mechanisms governing these PG-degrading enzymes remain poorly understood. Here, we unveil a novel and widespread control mechanism of lytic transglycosylases (LTs), a type of autolysin responsible for breaking down PG glycan chains. Specifically, we show that LD-crosslinks within the PG sacculus act as an inhibitor of LT activity. Moreover, we demonstrate that this regulation controls the release of immunogenic PG fragments and provides resistance against predatory LTs of both bacterial and viral origin. Our findings address a critical gap in understanding the physiological role of the LD-crosslinking mode in PG homeostasis, highlighting how bacteria can enhance their resilience against environmental threats, including phage attacks, through a single structural PG modification.

Bacteria are almost universally encased by a cell wall that is essential for their structural integrity. The main component of the cell wall is the peptidoglycan (PG), also known as murein, a net-like heteropolymer of glycan strands composed of repeats of the disaccharide *N*-acetylmuramic acid (MurNAc) and *N*-acetylglucosamine (GlcNAc), crosslinked by short peptides[1]. Because the so-called PG sacculus envelopes the cytoplasmic membrane and alters the interface with the surroundings, its chemical structure must be dynamically altered during growth and in response to the environmental changes[2,3].

Growth of PG relies on the coordinated action of synthetic and degradative enzymes. Synthetic enzymes include transglycosylases and transpeptidases, which polymerize and crosslink PG chains, respectively[4]. Among these, PG crosslinking is primarily mediated by Penicillin Binding Proteins (PBPs). However, in many bacteria, there are alternative transpeptidases known as LD-transpeptidases (LDTs)[5]. A key distinction between these two types of crosslinks lies in the

mechanisms of their formation: PBPs form new peptide bonds connecting the fourth D-Ala of the donor muropeptide and the D-chiral center of the diamino acid in the third position of an adjacent acceptor muropeptide, resulting in a 4,3- or DD-crosslink, while LDTs connect the L- and D-stereocenters of two adjacent diamino acid residues, generating a 3,3- or LD-crosslink[5]. Both crosslink types serve to strengthen the cell wall[6,7]. However, LDTs have the unique ability to form linkages not only within PG but also between PG and outer membrane proteins[8–10]. Although LDTs are not typically essential for bacterial survival, they play significant roles in various processes, including incorporation of non-canonical D-amino acids into PG[2,11], beta-lactam resistance[12–15] and preservation of cell envelope integrity during failure of LPS translocation[13]. Despite these known roles, the overall physiological function of LDTs remains unclear[5].

PG degrading enzymes, known as autolysins, cleave covalent bonds within the existing sacculus, enabling the insertion of newly

[1]Department of Molecular Biology, Umeå University, Umeå, Sweden. [2]Department of Microbiology, Cornell University, Ithaca, New York, USA. [3]Weill Institute for Cell and Molecular Biology, Cornell University, Ithaca, New York, USA. [4]Cornell Institute of Host-Microbe Interactions and Disease, Cornell University, Ithaca, New York, USA. [5]Umeå Center for Microbial Research (UCMR), Umeå University, Umeå, Sweden. [6]The Laboratory for Molecular Infection Medicine Sweden (MIMS), Umeå, Sweden. [7]Science for Life Laboratory (SciLifeLab), Umeå University, Umeå, Sweden. [8]Present address: Instituto de Bioquímica Vegetal y Fotosíntesis, Consejo Superior de Investigaciones Científicas and Universidad de Sevilla, Seville, Spain. [9]Present address: Department of Microbiology, Blavatnik Institute, Harvard Medical School, Boston, MA, USA. [10]These authors contributed equally: Laura Alvarez, Sara B. Hernandez. ✉e-mail: felipe.cava@umu.se

synthesized material. However, these activities can be a double-edged sword: while indispensable for bacterial growth, they need to be carefully controlled to prevent cell lysis[16,17].

The extent to which various classes of autolysins contribute to the expansion of the sacculus, cell division, and overall vegetative growth, as well as the mechanisms and regulatory processes involved, are subject of ongoing research. Autolysins are categorized based on the specific bonds they target within PG and are further differentiated by their unique catalytic domains[17,18]. Endopeptidases break the internal bonds, including crosslinks, within the peptide stem, while carboxypeptidases cut the terminal bonds. Amidases split the amide bond that connects the glycan strand to the stem peptide, specifically between MurNAc and L-alanine residues. Glycosidases, such as lysozyme or lytic transglycosylases (LTs), target the glycan strands themselves.

Lysozymes, also known as N-acetylmuramidases or muramidases, and LTs both act on the same β−1,4-glycosidic bond, albeit through distinct mechanisms. Lysozymes hydrolyze this bond, resulting in a product with a non-cyclic terminal reducing MurNAc residue. In contrast, LTs, which are not hydrolases, cleave the bond between MurNAc and GlcNAc, leading to the formation of anhydro-MurNAc capped products, also known as anhydromuropeptides[19,20]. LTs are broadly conserved in bacteria and can be categorized as exolytic if they attack glycosidic bonds located at the termini of the glycan chains and release soluble anhydromuropeptides, or endolytic if they cleave non-terminal bonds within the glycan chain[21,22].

LT enzymes are generally implicated in cell growth and division[23–25] and in the assembly of transenvelope complexes such as the type VI secretion system[26]. They are also involved in the release of extracellular PG fragments that serve as signals in adaptive responses such as beta-lactam resistance, and in bacteria-host interactions[27,28]. Although the structural and biochemical properties of LTs have been extensively studied, their regulation remains poorly understood.

Here, we used *Vibrio cholerae* as a Gram-negative model organism to study the changes that PG undergoes amidst environmental shifts. Our research has uncovered a widespread control mechanism: LD-crosslinks within the PG sacculus serve as inhibitors of the activity of LTs. PG analysis under different conditions alongside the use of *ldt* mutants that, while lacking LD-crosslinks, maintain the total crosslinking stable due to an increase in DD-crosslinks, underscored the superior inhibitory influence of LD-crosslinks compared to DD-crosslinks on the activity of LTs. Through various in vitro and in vivo approaches, we established that this structural interference selectively impacts LTs, while sparing lysozymes. We also discerned that the degree of LD-crosslinking is instrumental in regulating the dispersal of immunogenic PG fragments and provides a defense mechanism against LTs originating from both bacterial and viral origins. The observed fluctuations in LD-crosslinking under various conditions suggest that this mode of crosslinking is modulated by environmental factors, rather than being a direct countermeasure to LT-induced stress. This discovery highlights the physiological importance of LD-crosslinking in PG homeostasis and illustrates how bacteria can enhance their defenses against ecological menaces, such as phage incursions, by modulating their PG crosslinking strategy.

## Results

### Peptidoglycan profile screening reveals a correlation between LD-crosslinking and anhydromuropeptide levels

To understand how the cell wall adapts to different environmental challenges, we analyzed the PG composition of the bacterial pathogen *Vibrio cholerae* across more than a hundred physiologically relevant conditions (Fig. 1a, Supplementary Figs. 1, 2 and Supplementary Data 1). These chromatographic profiles provide detailed insights into the relative abundance and variations of each individual PG

component, known as muropeptides, across diverse culture conditions. Upon examining this PG dataset, we noticed a distinct inverse correlation in the levels of two major PG properties: LD-crosslinking and anhydromuropeptides, the chain termini generated by LTs (Fig. 1b, Supplementary Fig. 2). *V. cholerae* grown in minimal media with different carbon sources exhibited elevated LD-crosslinking and concomitant decreased anhydromuropeptide levels, while samples with the lowest LD-crosslinking had the highest anhydromuropeptide levels (Fig. 1c, Supplementary Fig. 3).

To further investigate the intriguing relationship between LD-crosslinking and LT activity in vivo, we used copper, known to inhibit LDTs of *Escherichia coli*[14]. We first sought to validate the inhibitory effect of copper on *V. cholerae* LDTs in vitro by testing its effect on purified LdtA, the primary LDT in *V. cholerae*[11] (Supplementary Fig. 4a). As in *E. coli*, *V. cholerae*'s LDT activity was inhibited by copper, showing full inhibition at concentrations that did not affect growth and morphology in LB medium (Supplementary Fig. 4b–e). No other metal salt we tested inhibited LDT activity both in vitro (Supplementary Fig. 4f) and in vivo (Supplementary Fig. 4g). Analysis of *V. cholerae* LD-crosslink (LDT activity) and anhydromuropeptide (LT activity) levels across varying concentrations of CuSO₄ in both minimal and rich media confirmed the concentration-dependent inverse correlation observed in our original PG screen (Fig. 1d, Supplementary Fig. 4h). The presence of CuSO₄ did not affect the activity of purified *V. cholerae* LTs (Supplementary Fig. 5), ruling out the possibility that the observed increase in anhydromuropeptide levels was a result of copper activating LTs.

Our results so far suggested that an increase in anhydromuropeptide levels is attributed to a reduction in LD-crosslinking. Supporting this idea, a *V. cholerae* Δ*ldt* mutant, which completely lacks LD-crosslinked muropeptides, also showed elevated levels of anhydromuropeptides compared to the wild-type strain, while complementation with LdtA greatly increased LD-crosslinking levels and reduced anhydromuropeptide content, especially in MM (Fig. 1e, Supplementary Fig. 6a, b). The exposure of the Δ*ldt* mutant to CuSO₄ did not alter the anhydromuropeptide levels, confirming that LTs are not affected by copper (Supplementary Fig. 6c). Further, despite comparable morphology and growth, this mutant exhibited a lower PG density than the wild-type strain, indicating increased PG degradation due to elevated LT activity (Supplementary Fig. 6b, d, e). Together, these findings suggest that a decrease in LDT activity positively impacts anhydromuropeptide levels in the PG, independent of the presence of copper. This effect is unidirectional, as LT mutants (i.e., *mltG* and *slt70*) with reduced anhydromuropeptide levels did not have an increase in LD-crosslinking (Fig. 1e, Supplementary Fig. 7). Thus, both our in vitro and in vivo results support the notion that LD-crosslinking interferes with LT activity, but not vice versa.

Importantly, this phenomenon is conserved across diverse bacterial species. Multiple LDT-encoding bacteria exhibited the same consistent pattern where copper-mediated inhibition of LD-crosslinking was accompanied by an increase in anhydromuropeptides (Fig. 1f, Supplementary Fig. 8a). These findings are further bolstered by previously reported data from other bacterial species presenting varying levels of LD-crosslinks due to mutations in their LDT enzymes or different growth states[13,29–31] (Supplementary Fig. 8b,c).

### LD-crosslinking in peptidoglycan hinders lytic transglycosylase activity

To directly test our hypothesis that LD-crosslinks repress LTs' autolytic activity, we conducted in vitro assays using a panel of purified LTs on PG substrates with varying LD-crosslink levels. We prepared these substrates by purifying *V. cholerae* Δ*ldt* sacculi (lacking LD-crosslinks) and incubating them with purified LdtA enzyme for different time periods (Fig. 2a), yielding PG samples containing increasing degrees of

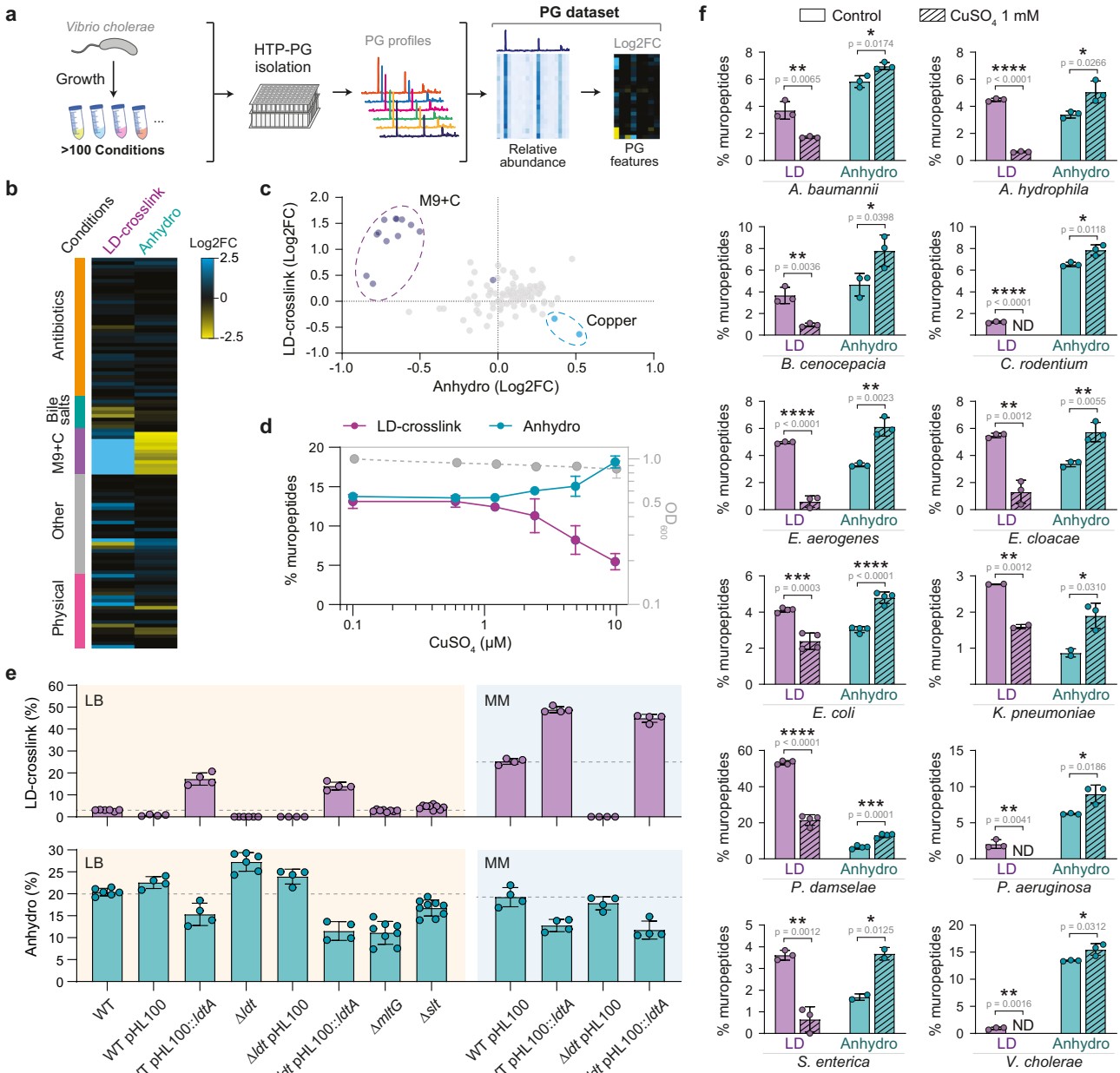

**Fig. 1 | Inverse correlation between LD-crosslinking and anhydromuropeptide levels. a** Overview of peptidoglycan profiling of the *V. cholerae* wild-type strain grown under different conditions. Peak abundances and main peptidoglycan (PG) features are calculated from the chromatographic profiles, and used for sample comparisons, studying correlations and exploration of PG dynamics. **b** Heatmap representing the Log2FC of LD-crosslink and anhydromuropeptide levels. The tested conditions are oriented along the y-axis, ordered by stress or the physiological condition (see Supplementary Data 1). **c** Scatter plot representing the Log2FC of anhydromuropeptide versus LD-crosslink levels, showing an inverse correlation between these features. PG analyses were performed in triplicate. Data are presented as mean values +/- standard deviation. M9 + C: minimal medium M9 supplemented with different carbon sources. **d** Concentration-dependent effect of copper on the levels of LD-crosslinking and anhydromuropeptides in the PG of *V. cholerae* grown in M9 medium with 0.4% (w/v) glucose (MM). **e** Relative levels of LD-crosslink and anhydromuropeptides in *V. cholerae* wild-type (WT),

mutant and complemented strains grown in LB and MM. PG analyses were performed in quadruplicate (or more replicates if needed). Data are presented as mean values +/- standard deviation. **f** Copper inhibits LD-crosslinking and promotes anhydromuropeptide accumulation in the PG of a variety of bacterial species grown in media supplemented with 1 mM CuSO₄ (*Acinetobacter baumannii, Aeromonas hydrophila, Burkholderia cenocepacia, Citrobacter rodentium, Enterobacter aerogenes, Enterobacter cloacae, Escherichia coli, Klebsiella pneumoniae, Photobacterium damselae, Pseudomonas aeruginosa, Salmonella enterica* and *Vibrio cholerae*). All bacteria were grown in LB, except for *P. damselae* which was grown in TSB medium. PG analyses were performed in triplicate. Data are presented as mean values +/- standard deviation. Statistical significance was determined using unpaired t-tests, with an alpha level of 0.05. Two-tailed *p*-values are reported. *$p < 0.05$; **$p < 0.01$; ***$p < 0.001$; ****$p < 0.0001$. ND: not detected. Source data are provided as a Source Data file.

LD-crosslinking (ranging from 0% to 17.5%). These samples were then utilized as substrates for in vitro reactions using purified *V. cholerae* LTs: MltA, MltB, MltD, MltG, and Slt70. We consistently observed a significant negative correlation between LD-crosslink levels and LT

activity in all cases (Fig. 2b). Some LTs, such as MltB or Slt70, appeared to be more sensitive to the presence of LD-crosslinks, indicating this crosslinking mode may be more effective at inhibiting specific LT enzymes in the cell.

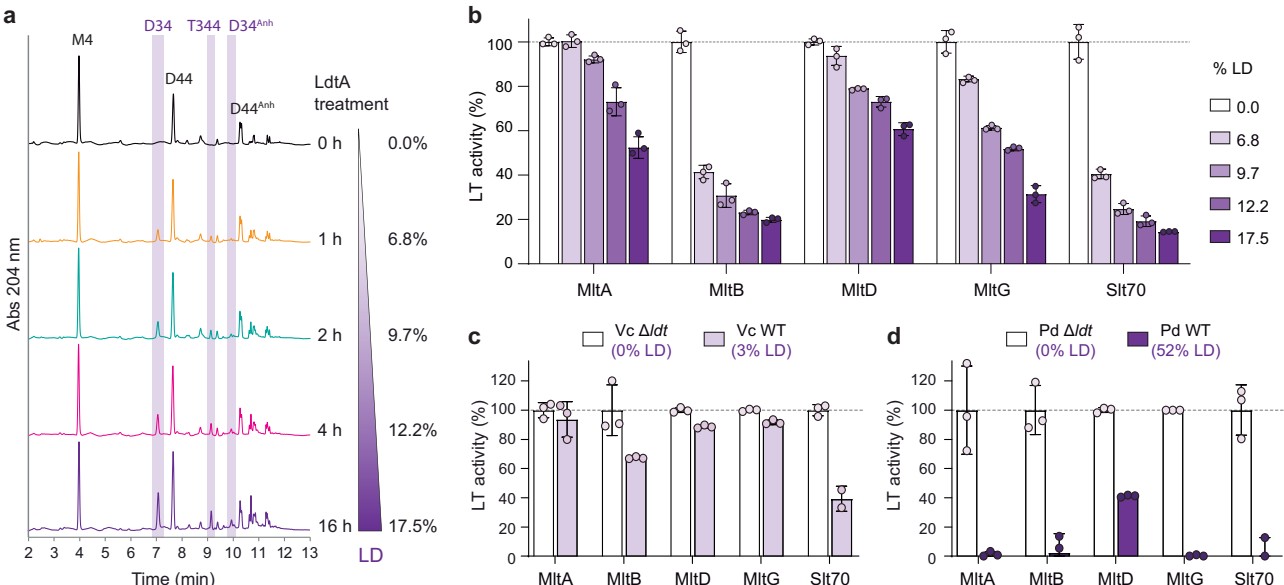

**Fig. 2 | Lytic transglycosylase activity is reduced on LD-crosslinked peptidoglycan. a** Representative chromatograms of purified *V. cholerae* Δ*ldt* mutant sacculi treated with LdtA for varying times. The percentage of LD-crosslinking is indicated on the right. **b** Activity of *V. cholerae* LTs (MltA, MltB, MltD, MltG and Slt70) on substrates with different LD-crosslinking levels relative to the sacculi substrate with 0% LD-crosslinks. **c** Relative activity of *V. cholerae* LTs (MltA, MltB, MltD, MltG and Slt70) on *V. cholerae* wild-type (Vc WT) or Δ*ldt* mutant (Vc Δ*ldt*) sacculi. Activity is calculated relative to the Δ*ldt* sacculi substrate, with 0% LD-crosslinks. **d** Relative activity of *V. cholerae* LTs (MltA, MltB, MltD, MltG and Slt70) on *Photobacterium damselae* wild-type (Pd WT) or Δ*ldt* mutant (Pd Δ*ldt*) sacculi. Activity is calculated relative to the Δ*ldt* sacculi substrate, with 0% LD-crosslinks. All in vitro assays were performed in triplicate. Data are presented as mean values +/- standard deviation. Source data are provided as a Source Data file.

We obtained additional evidence from further LT assays, using sacculi from both wild-type and Δ*ldt* mutant strains of *V. cholerae* and *Photobacterium damselae* (Supplementary Fig. 9a,b). While *V. cholerae* grown in LB exhibits approximately 3% LD-crosslinks in its cell wall, its close relative *P. damselae* contains a significantly higher proportion, reaching up to 52%. All LTs showed significantly higher activity on Δ*ldt* sacculi compared to wild-type, with activity levels dropping drastically (60–100%) when using wild-type sacculi from *P. damselae* as the substrate (Fig. 2c,d). These findings were not specific to *V. cholerae's* LTs as they were recapitulated using *E. coli's* Slt70 (Supplementary Fig. 9c,d). It is important to note that while LD-crosslinks are reduced in the peptidoglycan of the Δ*ldt* mutants, their overall crosslinking levels remain comparable to those of the wild-type strains (Supplementary Fig. 9b). This emphasizes the significance of the crosslink mode over the total degree of crosslinking in influencing LT activity. Moreover, this regulatory effect was also evident in vivo, as indicated by a notable increase in soluble anhydromuropeptides released into the supernatants of the Δ*ldt* mutants compared to those of the wild-type strains (Supplementary Fig. 9e).

Taken together, these findings illustrate that LD-crosslinks, in contrast to DD-crosslinks or crosslinking in general, restrict LT enzymatic activity, thereby influencing the abundance of extracellular PG fragments.

## LD-crosslinked peptidoglycan suppresses bacterial and phage lytic transglycosylase activity

Many predatory PG degrading systems of both bacterial and phage origin rely on LT activity (Fig. 3a). Thus, we reasoned that LD-transpeptidation may play a protective role against the action of these enzymes. To test this hypothesis, we purified two exogenous LTs: the type VI secretion system (T6SS) Tse4 effector of *Acinetobacter baumannii*[32] and the LaL endolysin from the lambda bacteriophage[33]. We then performed in vitro digestions of sacculi with varying degrees of LD-crosslinking. Similar to *V. cholerae* LTs, Tse4 and LaL in vitro activities were strongly negatively affected by LD-crosslinking, with a

50% reduction in their in vitro activity (Fig. 3b, Supplementary Fig. 10). To test whether LD-crosslinks could also interfere with the activity of other PG degrading enzymes, we tested chicken egg white lysozyme and *Streptomyces globisporus* mutanolysin (i.e., muramidase)[34]. Remarkably, both enzymes retained their full activity regardless of the degree of LD-crosslinking. Lysozymes and muramidases cleave the same β-1,4-glycosidic bonds between MurNAc and GlcNAc as LTs; however, they do it in a hydrolytic fashion (forming a reducing end at the MurNAc)[35]. Therefore, these results indicate that LD-crosslinks specifically negatively affect the activity of LTs, but not of lysozymes.

To investigate if LD-crosslinks could also provide protection in vivo, we used *E. coli* as a model in bacteriophage infection assays. We compared the lytic effect of the lambda phage on *E. coli* cultures that differed in the amount of LD-crosslinking in their PG. To increase the LD-crosslink levels, we overexpressed the *E. coli* LDT LdtE[6,36] from an inducible promoter and used non-induced and empty vector as controls (Supplementary Fig. 11a,b,c). Overexpression of LdtE resulted in a marked reduction of lysis in infected cultures (Fig. 3c). Similarly, LdtE expression reduced the number of plaques by more than 1.5 orders of magnitude in an agar overlay assay (Fig. 3d,e). Confirming our hypothesis, these results recapitulated only when *E. coli* was infected with other LT-encoding phages, such as P2, but not when using phages encoding lysozyme or endopeptidase-like endolysins (i.e., P1, T4 or T5) (Fig. 3e, Supplementary Fig. 12). Together, these results demonstrate that increasing the degree of LD-crosslinking improves the resilience of the cell wall against predatory LTs of both bacterial and phage origin.

## Discussion

The PG cell wall is crucial for the survival of most bacteria, making it imperative to tightly regulate the enzymes responsible for its degradation to avoid cell lysis. However, the mechanisms governing this regulation are not fully understood. Using *V. cholerae*, we examined how the bacterial cell wall adapts to different environments. We

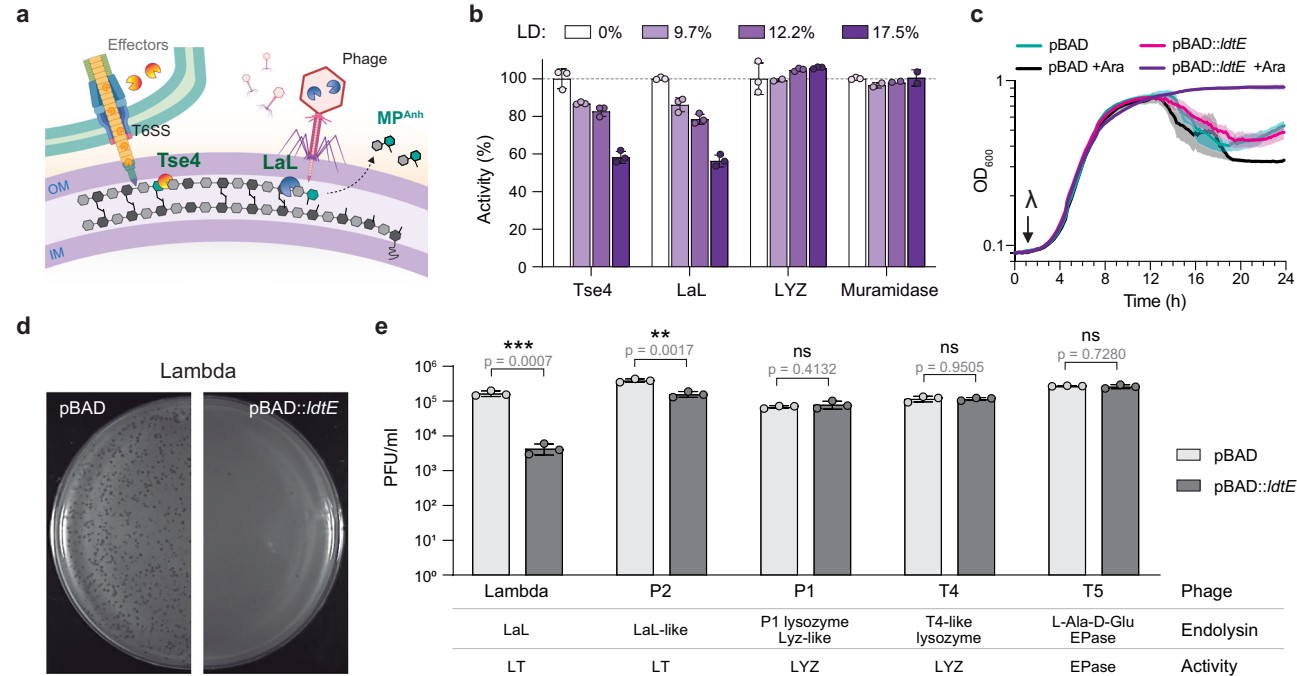

**Fig. 3 | Effect of LD-crosslinks on peptidoglycan degrading enzymes.**
**a** Schematics showing the activity of exogenous lytic transglycosylases on the cell wall. Panel created with BioRender.com, released under a Creative Commons Attribution-NonCommercial-NoDerivs 4.0 International license (https://creativecommons.org/licenses/by-nc-nd/4.0/deed.en). **b** Activity of different PG-degrading enzymes on substrate with indicated LD-crosslinking levels. Activity is calculated relative to sacculi substrate with 0% LD-crosslinks. Enzymes: Tse4 effector of *A. baumannii*, bacteriophage lambda endolysin (LaL), chicken egg white lysozyme (LYZ) and mutanolysin from *Streptomyces globisporus* (Muramidase). In vitro assays were performed in triplicate. Data are presented as mean values +/- standard deviation. **c** Growth curves of *E. coli* JM109 with empty pBAD33 (pBAD) plasmid or pBAD::*ldtE*, grown with and without inducer (arabinose 0.2% (w/v),

+Ara), infected at time zero with the same amount (~$10^4$ PFUs) of lambda phage particles ($\lambda$). **d** Representative plates showing increased resistance of the LDT-overexpressing strain to lambda phage plaque formation. LB agar plates are supplemented with 20 µg/ml chloramphenicol, 10 mM $MgSO_4$, and 0.2% (w/v) arabinose. **e** Quantification of PFU/ml produced by different phages encoding endolysins with lytic transglycosylase (LT), lysozyme (LYZ) or endopeptidase (EPase) activities on *E. coli* JM109 carrying empty pBAD or pBAD::*ldtE* at 24 h post-infection. Infection assays were performed in triplicate. Data are presented as mean values +/- standard deviation. Statistical significance was determined using an unpaired t-test, corrected for multiple comparisons by the Holm-Sidak method, with an alpha level of 0.05. Two-tailed *p*-values are reported. **$p < 0.01$; ***$p < 0.001$; ns: not significant. Source data are provided as a Source Data file.

uncovered a widely conserved mechanism that controls the activity of LTs, a diverse family of cell wall autolysins (Fig. 4).

While *Bacillus* species employ specific spatial protein localization, protein-protein interactions, and small molecules to regulate the activity of their LTs during spore germination[24], such sophisticated mechanisms are uncommon among most bacteria. Although alternative regulatory mechanisms for LTs exist, they are often poorly conserved[37–40]. Additionally, certain pathogenic species modify their PG sugars through O-acetylation, which provides protection against LTs but also confers resistance to other glycan-degrading enzymes like lysozymes[41,42]. In our study, we discovered that LD-crosslinking specifically inhibits LTs from both bacterial and viral sources. LD-crosslinks are synthesized by LDTs, which are widely conserved enzymes found in many Gram-negative and Gram-positive taxa[6,9,11,15,43,44]. Consequently, this regulatory mechanism is poised to have wide-ranging biological implications for bacteria.

How do LD-crosslinks hinder LT activity? Our data suggest that these crosslinks restrict LTs' ability to access their substrate. This idea is supported by previous studies demonstrating that widely conserved LTs like MltC feature an elongated peptidoglycan-binding groove, which preferentially recognizes long uncrosslinked peptidoglycan strands[45]. While this mechanism restricts LTs from degrading LD-crosslinked regions, it still requires high crosslinking densities to be effective. However, our results indicate that even low levels of LD-crosslinking can significantly inhibit LT activity (Fig. 2c). Exolytic LT enzymes typically trim the PG starting from the chain termini, which are often enriched in crosslinked muropeptides. Specifically, the ratio

of dimers or trimers to monomers is higher for anhydromuropeptides compared to their non-anhydro counterparts[46]. Considering that the average length of PG chains in bacteria varies widely among species, ranging from a few disaccharides per chain to several hundred[46], even a small number of LD-crosslinks could potentially impose significant structural constraints that effectively inhibit LT PG-degrading activity.

While previous studies suggested that crosslinked PG could influence the processivity of certain exolytic LTs[25,28,47–49], our data indicate that PG containing DD-crosslinks is degraded more efficiently than PG containing LD-crosslinks (Fig. 2). This is evidenced by the higher LT activity observed on PG with lower LD-crosslinking levels, even as DD-crosslinking compensates to maintain global crosslinking homeostasis (Supplementary Fig. 9b). We hypothesize that the shorter peptide bond of LD-crosslinks compared to DD-crosslinks[50] may impede the processivity of these enzymes along the PG chains. Interestingly, some LT enzymes appear to be more strongly affected by LD-crosslinking than others, suggesting that LTs may exhibit varying activity states at specific LD-crosslinking levels. Encoding LTs that are less sensitive to LD-inhibition might be particularly important for bacteria with naturally high levels of LD-crosslinked cell walls such as *Agrobacterium tumefaciens*, *Mycobacterium tuberculosis*, or *P. damselae*[51–53]. Future biochemical and structural studies should investigate the coevolution and functional diversification of LT enzymes to adapt to species-specific LD-crosslinking patterns.

This mechanism of LT control strengthens the association between LD-crosslink and beta-lactam resistance beyond the mere fortification of the cell wall by alternative PG crosslinking enzymes that

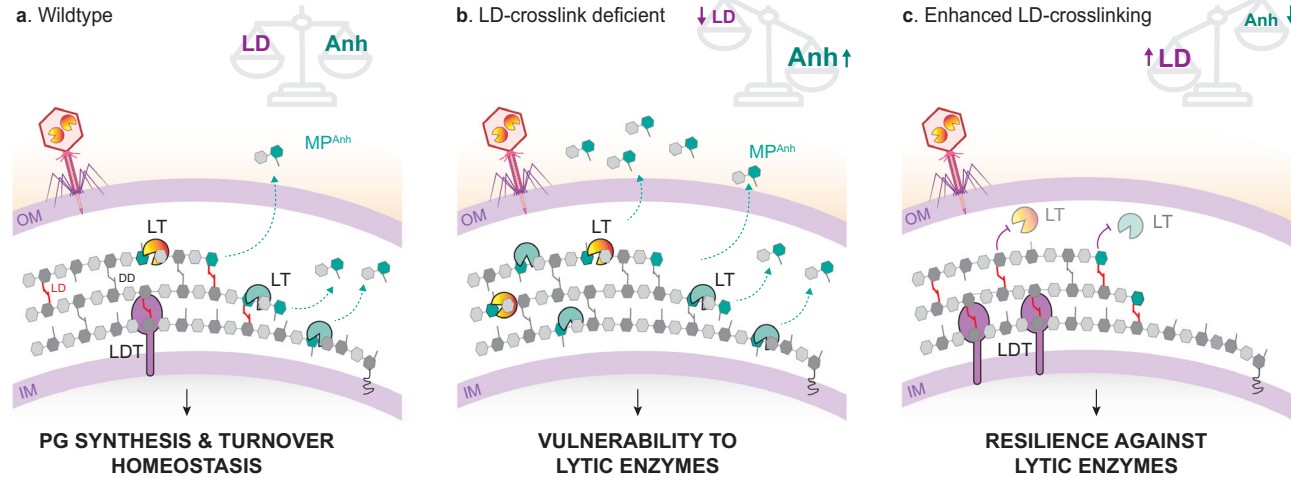

**a.** Wildtype

**b.** LD-crosslink deficient

**c.** Enhanced LD-crosslinking

**PG SYNTHESIS & TURNOVER HOMEOSTASIS**

**VULNERABILITY TO LYTIC ENZYMES**

**RESILIENCE AGAINST LYTIC ENZYMES**

**Fig. 4 | Model for LD-crosslink mediated control of lytic transglycosylase activities. a** PG synthesis and turnover are balanced in the wild-type bacteria. LTs from endogenous (autolysins) and exogenous origin (bacterial and viral effectors) hydrolyze the PG chains and liberate anhydromuropeptides. LD-crosslinked chain termini modulate the activity of these enzymes. **b** During LD-crosslink deficiency, the cell wall is more vulnerable both to autolysins and predatory lytic enzymes which can degrade it more efficiently. **c** Conversely, enhanced LDT activity leads to increased LD-crosslinking in the cell wall, inhibiting LTs and reinforcing the cell wall's resilience. Figure created with BioRender.com, released under a Creative Commons Attribution-NonCommercial-NoDerivs 4.0 International license (https://creativecommons.org/licenses/by-nc-nd/4.0/deed.en).

are less sensitive to the inhibitory action of beta-lactam antibiotics[7,12,15]. While replacing DD-crosslinks with the alternative LD-type when PBPs are inactivated by beta-lactams makes the PG more resistant to autolysis and favors cell stability, inhibition of LT activities by LD-crosslinking also influences the release of anhydromuropeptides (Supplementary Fig. 9e). A reduction in LD-crosslink levels activates LTs, resulting in increased extracellular anhydromuropeptide levels, potentially contributing to the upregulation of the inducible beta-lactamase AmpC in some bacterial species[54]. Moreover, given that extracellular PG fragments serve diverse signaling functions, including involvement in developmental transitions, innate immune responses, organ development, and symbiosis[27], the modulation of LDT activity could potentially affect these processes. For example, recent reports have indicated that LDTs are upregulated during *Salmonella* intracellular infection, indicating that increased LD-crosslinking in vivo may play a crucial role in evading the host immune system and facilitating the establishment of a persistent stage within eukaryotic cells[55]. Therefore, revealing the mechanisms and stimuli that govern LDTs activity is crucial for understanding how LTs are regulated and its subsequent effects on host-bacteria interactions, infection dynamics, and antibiotic resistance.

Our results further demonstrate that LD-crosslinked PG broadly inhibits both endogenous LTs as well as predatory LTs of both bacterial and phage origin (Fig. 3). This suggests that adjusting the level of LD-crosslinking could enhance the resilience of the cell wall to phage attack or interbacterial competition. Although, our results show that LD-crosslinking does not increase as a result of lambda phage infection (Supplementary Fig. 11d,e), the variation in the number (e.g., 2 in *V. cholerae*, 6 in *E. coli* and 14 in *A. tumefaciens*[56]) and regulatory mechanisms of LDTs across different species (LdtA$_{Vc}$ and LdtD$_{Ec}$ controlled by RpoS and Cpx, respectively[5]) makes it plausible that external LT attacks, such as those from specific phages, may provoke a defensive LD-crosslinking reaction in certain bacteria. This protective function aligns with previous research indicating that LDTs are induced by envelope stress[11,57,58] and can repair and reinforce the PG to compensate for structural imbalances caused by OM perturbations[13]. In this line, the predator *Bdellovibrio bacteriovorus* uses LDT enzymes to modify the PG of its prey during infection, establishing a stable, stress-resistant intracellular niche[59]. Future studies should reveal if LD-crosslinking mediated resistance to predatory LTs is a direct response

to the damage itself or it is rather predetermined by the environmental contexts controlling the LDTs' expression, and if predator-prey dynamics can evolve some predatory LTs to cleave LD-crosslinked PG.

Finally, although our PG chemical analysis established a clear inverse relationship between LD-crosslinking and anhydromuropeptide levels, certain exceptions indicate that we must also account for additional regulatory layers affecting the expression and activity of the LTs. For example, our prior studies have shown a decrease in LD-crosslinking in the presence of bile in *Salmonella*[29]. A similar pattern is observed in *V. cholerae*, yet without a corresponding rise in anhydromuropeptide levels, hinting at bile's possible inhibitory impact on LTs' function or production in this species. This points to the necessity for further investigation into the environmental factors governing LT regulation.

Overall, our study has revealed a fundamental mechanism for control of autolysin activity with broad implications in bacterial growth, morphogenesis, pathogenesis, antibiotic resistance, interbacterial competition and phage resistance. Given the widespread nature of this regulation and the importance of the cell wall as antibiotic target, our results can help in the identification of new points of vulnerability with potential clinical applications.

## Methods

### Bacterial strains and culture conditions
Bacterial strains are listed in Supplementary Table 1. All strains were grown under optimal conditions (media and temperature) recommended by the DSMZ or the ATCC bacterial collections, as indicated in the table.

Lysogeny broth (LB) containing 10 g/l NaCl or M9 minimal medium supplemented with 0.4% (w/v) glucose (MM) were used as the standard growth media. 1.5% (w/v) agar was used in solid plates. When required, antibiotics were added to the culture medium or plates at the following concentrations: streptomycin 200 μg/ml (added to *V. cholerae* cultures), ampicillin 100 μg/ml, kanamycin 50 μg/ml, and chloramphenicol 20 μg/ml. CuSO$_4$ was added to media at the indicated concentrations.

For complementation in *V. cholerae*, plasmid pHL100::*ldtA*[11] was introduced in the corresponding strains via electroporation. Expression of LdtA was induced with 1 mM IPTG from the $P_{lac}$ promoter. Empty pHL100 plasmid was used as control[11].

For the screening under different conditions, *V. cholerae* C6706 wild-type was grown overnight at 37 °C and diluted 1000-fold into 5 ml cultures, then incubated for 16 h under the conditions listed in Supplementary Data 1. Unless otherwise indicated, all cultures were agitated during growth.

## Plasmid construction

Plasmids (Supplementary Table 2) were constructed by using standard DNA cloning techniques. The oligonucleotides used for the construction are listed in Supplementary Table 3. The fidelity of the DNA regions generated by PCR amplification was confirmed by DNA sequencing. *E. coli* DH5α was used for plasmid construction, then plasmids were transferred to BL21 or JM109 strains for protein expression.

Plasmids used for protein production and purification are derivatives of pET plasmids pET15b, pET22b or pET28b (Novagen). Coding regions of the genes of interest were cloned into the multiple cloning site (MCS) of the pET vector with the 6xHis tag at the N- or C-terminus and expressed under the control of the $P_{lac}$ promoter. Residues corresponding to the signal peptide and transmembrane domain were removed from all membrane associated proteins to enhance expression and solubility of the protein.

The plasmid used for overexpression of *E. coli*'s LDT LdtE is a derivative of pBAD33[60], where expression of the gene is controlled by the $P_{BAD}$ promoter.

Genes under the control of $P_{lac}$ and $P_{BAD}$ promoters in pET and pBAD33 plasmids were induced with 0.5–1 mM isopropyl-β-D-1-thiolgalactopyranoside (IPTG) or 0.2% (w/v) L-arabinose, respectively.

## Fitness and growth measurements

To assess the fitness of *V. cholerae* grown under the different conditions or the effect of copper on diverse bacteria, overnight cultures were diluted 10-fold in LB and the optical density at 600 nm ($OD_{600}$) was measured using an Eon plate reader (Biotek, USA). Biotek Gen5 [v.08] software was used to collect OD measurements. Cultures were then subjected to serial 10-fold dilution and 100 μl aliquots of the $10^{-5}$ through $10^{-7}$ dilutions were inoculated onto agar plates. Then, the plates were incubated at the appropriate temperature for 16–24 h prior to CFU counting.

For continuous growth monitoring, at least three replicates per strain and condition tested were grown in two independent experiments in 200 μl medium in a 96-well plate inoculated 1:1000 from exponentially growing precultures. $OD_{600}$ was monitored in the plate reader at 10 min intervals, using the optimal growth temperature for the tested species. When needed, growth media (LB or MM) were supplemented with the indicated concentrations of $CuSO_4$.

## Microscopy

Bacteria were immobilized on LB pads containing 1% (w/v) agarose. Phase contrast microscopy was performed using a Zeiss Axio Imager Z2 microscope (Zeiss, Germany) equipped with a Plan-Apochromat 63X phase contrast objective lens and an ORCA-Flash 4.0 LT digital CMOS camera (Hamamatsu Photonics, Japan), using the Zeiss Zen Blue [v2.0.0.0] software. Image analysis and processing were performed using the Fiji/ImageJ [v1.53] software[61]. The MicrobeJ plugin was used for cell width and length measurements, using manually edited cell outlines as needed[62]. Microscopy images shown are representative of three biological replicates.

## High throughput PG isolation and digestion

PG preparation was performed as described[63]. Briefly, 500 μl bacterial cultures were pelleted in 96-well deep-well plates. After the supernatants were discarded, the pellets were resuspended with 50 μl of fresh medium and 25 μl of 5% (w/v) SDS were added, prior to autoclaving (15 min at 120 °C 1 atm). Samples were then transferred to 96-well filter plates (AcroPrep Advance 96-well 0.2 μm GHP membrane filter plates, 2 ml volume, catalog number 8282, PALL, USA). SDS was removed by several washes with 1 ml Milli-Q water, with centrifugation for 15 min at 5,250 *x g*. SDS-free sacculi samples were treated with 200 μl proteinase K (40 μg/ml in 100 mM TrisHCl pH 8.0, 30 min at 37 °C) for removal of the Braun's lipoprotein. After an additional wash with water, samples were digested overnight at 37 °C with 50 μl muramidase (100 μg/ml in water). Enzymatic reactions were performed directly on the filter, sealing the plates with aluminum foil, and using a humidity chamber to reduce evaporation. After muramidase digestion, soluble muropeptides were eluted into a new 96-well deep-well plate and reduced using $NaBH_4$ (10 mg/ml) in borate buffer (0.5 M pH 9.0) for 30 min at room temperature. Finally, the pH of the samples was adjusted to 3.5 using 25% (v/v) orthophosphoric acid and the samples were stored frozen at −20 °C until analysis by liquid chromatography.

## Data acquisition by liquid chromatography and MS analysis

PG samples were analyzed as previously described[64]. Chromatographic analyses of muropeptides were performed by Ultra Performance Liquid Chromatography (UPLC) using Empower 3.6 software (Waters, USA) on an UPLC system (Waters, USA) equipped with a trapping cartridge precolumn (SecurityGuard ULTRA Cartridge UPLC C18 2.1 mm, catalog number: AJ0-8768, Phenomenex, USA) and an analytical column (BEH C18 column, 130 Å, 1.7 μm, 2.1 mm by 150 mm, catalog number: AJ0-8768, Waters, USA). Muropeptides were separated using a linear gradient from buffer A (Water + 0.1 % (v/v) formic acid) to buffer B (Acetonitrile + 0.1 % (v/v) formic acid) over 15 min with a flowrate of 0.25 ml/min. Muropeptides were detected by measuring UV absorbance at 204 nm. Chromatograms shown are representative of three biological replicates.

Muropeptide identity was confirmed by MS and MS/MS analysis, using a Xevo G2-XS Q-tof system (Waters, USA). The instrument was operated in positive ionization mode. Detection of muropeptides was done by the data-independent $MS^E$ mode (where exact mass data for every detectable component and its fragment ions are recorded) using the following parameters: capillary voltage 3.0 kV, source temperature 120 °C, desolvation temperature 350 °C, sample cone voltage 40 V, cone gas flow 100 l/h, desolvation gas flow 500 l/h, and collision energies (CE) of 6 eV (low CE) and 15–40 eV (ramped, high CE). The scan time was 0.25 s, with a mass range of *m/z* 100–2000. Data acquisition and processing was performed using the UNIFI 1.8.1 software (Waters, USA).

## PG chromatographic analysis and data transformation

Collected UPLC chromatographic data were analyzed using the PG-metrics pipeline in MATLAB R2023b[65]. In brief, raw data were preprocessed by trimming out irrelevant segments and baseline correction was done by subtracting a synthetically created baseline. Trimmed and baseline-corrected datasets were then aligned in segments using the COW algorithm by selecting appropriate reference samples and combinations of the segment length and slack parameters. The peak area of aligned chromatograms was calculated using the trapezoidal integration approach, where the boundaries of the integration were manually selected.

The relative area of each muropeptide was calculated by dividing its peak area by the total area of the chromatogram. The different PG features were calculated as described previously[64,66] and are described in Supplementary Data 1. Log2 fold change (Log2FC) was calculated relative to growth on LB 10 g/l NaCl at 37 °C.

## Extracellular soluble MPs analysis

Sample preparation of extracellular soluble PG samples was performed as described[67]. 1 ml of bacterial cultures were harvested by centrifugation (1,500 *x g* for 20 min at 4 °C). The supernatants were

filtered through 0.2 μm pore size filters, boiled for 15 min, and precipitated proteins were removed by centrifugation (22,000 $x\,g$ for 15 min). Finally, sample pH was adjusted to pH 3.5 with 25% (v/v) orthophosphoric acid. When needed, samples were diluted or concentrated using a SpeedVac system.

Muropeptides were detected and characterized by MS and MS/MS analysis as described above. An in-house compound library built in UNIFI was used for identification. Quantification was done by integrating peak areas from extracted ion chromatograms (EICs) of the corresponding $m/z$ value of each muropeptide. Soluble muropeptide analyses were performed in biological triplicates and means were compared by unpaired t-test.

### Protein expression and purification

For protein purification, large culture volumes were used (250–1000 ml). *E. coli* BL21 cells carrying pET derivative plasmids were grown at 37 °C in LB media containing ampicillin (100 μg/ml) and 0.4% glucose (w/v) until an $OD_{600}$ of 0.4 was reached. Then, expression of the His-tagged proteins was induced by addition of 0.5–1 mM IPTG and cultures were incubated for 2 h at 37 °C. In cases when the protein induction was poor, induced cultures were incubated overnight at 16 °C. Harvested cells were resuspended in PBS (phosphate-buffered saline solution, pH 7.4) and cell crude extracts were disrupted twice in a French press followed by centrifugation at 20,000 $x\,g$ for 30 min at 4 °C. Proteins were purified from the soluble fraction by affinity chromatography using Ni-NTA agarose (catalog number: 30210, QIAGEN, Germany) previously equilibrated with PBS. Protein binding was done in PBS with 20 mM imidazole for 2 h at 4 °C, using a rotatory wheel for gentle agitation. After washing with PBS with 20 mM imidazole, proteins were eluted in PBS with 500 mM imidazole. Finally, imidazole was removed by dialysis against PBS overnight at 4 °C using a Spectra/Por® molecular porous membrane tubing (6–8 kDa cutoff) (catalog number: 132650, SpectrumLabs, USA). Purified proteins were visualized by SDS-PAGE and quantified by Bio-Rad Protein Assay (catalog number: 500–0006, Bio-Rad, USA) using BSA as standard. The proteins were either stored at 4 °C for immediate use, or at −80 °C after addition of 10% (v/v) glycerol.

### In vitro enzymatic reactions

Purified M4 was used as substrate in LDT assays. For its purification, *V. cholerae* saccculi were digested with muramidase (100 μg/ml) for 16 h at 37 °C and peaks were separated and collected by HPLC using the separation conditions described above. Collected peaks were lyophilized, dissolved in MilliQ water and stored at −20 °C. The concentration was determined by running a small aliquot in the UPLC as described above and comparing the peak area to a standard.

Purified saccculi were used as substrate in LT in vitro enzymatic reactions. For isolation, 500 ml cultures of *V. cholerae, E. coli* BW25113 or *P. damselae* wild-type or Δ*ldt* strains were grown at their optimal growth conditions. Saccculi were purified following standard PG isolation protocols[64,68]. Briefly, cells were harvested, then resuspended and boiled in 5% (w/v) SDS for 1 h. Saccculi were repeatedly washed by ultracentrifugation (150,000 $x\,g$ for 10 min at 20 °C) with MilliQ water until SDS was totally removed. Samples were treated with 20 μg/ml Proteinase K (1 h at 37 °C) for Braun's lipoprotein removal. After heat inactivation by boiling for 10 min with 1% (w/v) SDS, saccculi were washed again by ultracentrifugation and finally resuspended in MilliQ water. The saccculi concentration was assessed by digesting a small aliquot with muramidase and analyzing it by UPLC as described above.

LDT assays were performed in 25 μl reactions with 50 mM TrisHCl pH 7.5, 100 mM NaCl buffer containing 5 μg of purified M4 and the indicated amounts of metal salts. 10 μg of LdtA were added and reactions were incubated at 37 °C for 16 h. All enzymatic reactions were stopped by boiling the samples for 5 min, followed by centrifugation at 22,000 $x\,g$ for 15 min to remove precipitated protein. The pH of the supernatant containing the reaction products was adjusted to 3–4 with 25% (v/v) orthophosphoric acid and then the samples were injected in a UPLC for analysis. Relative LDT activity was calculated as the % of D34 produced compared to the amount of D34 produced in reactions not supplemented with metal salts (100%).

For the preparation of saccculi with varying degrees of LD-cross-linking, *V. cholerae* Δ*ldt* saccculi were treated with LdtA for 0, 1, 2, 4 or 16 h as described above. The relative amount of LD-crosslinks in the resulting saccculi was determined by injection of muramidase-treated, reduced and pH adjusted samples in a UPLC. The LdtA-treated saccculi were heat-inactivated, centrifuged, resuspended with 50 μl MilliQ water and used as substrate for subsequent LT in vitro assays.

In vitro assays to test the degrading activity of the different LTs (endogenous *V. cholerae* LTs, *E. coli* Slt70, *A. baumannii* Tse4 and LaL) were performed in 25 μl reactions with 50 mM TrisHCl pH 7.5, 100 mM NaCl buffer containing 5 μg of purified saccculi as substrate and the indicated amounts of metal salts. 10 μg of purified LTs were added and reactions were incubated at 37 °C for 16 h. The enzymatic reactions were heat-inactivated for 10 min at 100 °C. Soluble (released muropeptides and fragments) and pellet (intact saccculi) fractions were separated by centrifugation at 22,000 $x\,g$ for 15 min. 75 μl of water was added to the soluble fractions, and their pH was adjusted to 3–4 with 25% (v/v) orthophosphoric acid. Pellet fractions were resuspended in 50 μl of water. 2 μl of muramidase (1 mg/ml) was added and samples were further incubated at 37 °C for 16 h. After heat-inactivation for 10 min at 100 °C, soluble products were reduced with sodium borohydride, and their pH adjusted as indicated above. Finally, both fractions were injected in the UPLC for analysis. Exolytic LTs release soluble anhydromuropeptides, while endolytic LTs modify the relative abundance of anhydromuropeptides in the remaining insoluble fraction. Relative exolytic LT activity was calculated as % of released muropeptides compared to the released muropeptides in reactions not supplemented with metal salts. When different substrate saccculi were used, activity was calculated relative to the saccculi substrate with 0% LD-crosslinks. All reactions were performed in triplicate and statistical analysis was performed using two-tailed unpaired t-test.

### Phage infection assays

The genomes of all phages within the Basel *E. coli*-phage collection[69] were analyzed and the encoded endolysins were then categorized based on homology and functional domains information from UniProt[70] and the Conserved Domain Database[71]. Data are presented in Supplementary Table 4. Phages lambda, P1, P2, T4 and T5, each representing a distinct endolysin type, were selected and used in infection assays. Lambda and P2 encode lambda-lysozyme endolysins; P1 encodes a P1 lysozyme Lyz-like protein; T4 encodes a bacteriophage T4-like lysozyme; and T5 encodes an L-alanyl-D-glutamate endopeptidase.

The stock of phage used for bacterial infections was prepared by bacteriophage propagation in liquid broth[72] by infecting cultures of the *E. coli* K-12-derived laboratory strain C600 (grown at 37 °C in LB medium supplemented with 10 mM $MgSO_4$) with the different phages. After overnight incubation, the cell lysates were centrifuged to remove cell debris (1500 $x\,g$ for 20 min). The phage lysates were cleaned using 0.22 μm filters, chloroform was added, and the stocks were stored at 4 °C. Phage titration was determined by plaque assay[73] using the *E. coli* C600 strain.

Resistance to lysis was measured on the lambda phage sensitive *E. coli* JM109 overexpressing LdtE (pBAD::*ldtE*) or carrying the empty pBAD vector as control. Arabinose (0.2% (w/v) L-arabinose) was used for induction of the $P_{BAD}$ promoter. Lysis in liquid cultures was analyzed by monitoring the growth of *E. coli* JM109 strains (initial number of cells ~$10^6$) in LB supplemented with 20 μg/ml chloramphenicol and 10 mM $MgSO_4$ (and 0.2% (w/v) of arabinose when indicated) at 30 °C, after addition of around $10^4$ plaque-forming units (PFUs) of lambda

phage. The $OD_{600}$ of the cultures was measured at 10 min intervals using an Eon plate reader (Biotek, USA). Three replicates per strain and condition were tested and the experiment was repeated three times.

For plaque assays, 100 µl of diluted phage lysates were mixed with 200 µl of *E. coli* JM109 cultures prepared by 10-fold dilution of overnight cultures and grown to stationary phase ($OD_{600}$ ~1) at 37 °C in LB supplemented with 20 µg/ml chloramphenicol and 0.2% (w/v) arabinose. 10 mM $MgSO_4$ was added to the mixtures to facilitate phage adsorption and samples were incubated for 15 min at room temperature without agitation. These mixtures were added to 3 ml of molten top LB agar (0.35% (w/v)) supplemented with 10 mM $MgSO_4$ and overlaid on plates containing LB agar (1.5% (w/v)), 20 µg/ml chloramphenicol and 0.2% (w/v) arabinose. The plates were incubated 24 h at 30 °C for plaque formation. The number of PFUs produced on each *E. coli* strain was compared.

### Data analysis and representation
Heatmaps were generated in R v 4.3 using the ggplot2[74] and pheatmap (https://github.com/raivokolde/pheatmap) packages.

Prism 8.0 (GraphPad Software, USA) was used to plot and analyze numerical data.

Figures were created using Illustrator 24.0 (Adobe, USA). Figures 3a and 4 include content created with BioRender.com and are released under a Creative Commons Attribution-NonCommercial-NoDerivs 4.0 International license (https://creativecommons.org/licenses/by-nc-nd/4.0/deed.en).

### Statistics and reproducibility
Statistical significance was assessed using Prism 8.0 (GraphPad Software, USA). Two-tailed unpaired t-tests were used for statistical calculations. When needed, multiple comparisons were corrected using the Holm-Sidak method. A $p$ ($p$-value) $< 0.05$ was considered statistically significant (*$p < 0.05$; **$p < 0.01$; ***$p < 0.001$; and ****$p < 0.0001$).

Appropriate controls were used in all assays; control and experimental groups were done in isogenic strains. Assays were performed with at least three biological replicates unless otherwise indicated. No statistical method was used to predetermine the sample size. No data were excluded from the analyses. The experiments were not randomized. The investigators were not blinded to allocation during experiments and outcome assessment.

### Reporting summary
Further information on research design is available in the Nature Portfolio Reporting Summary linked to this article.

## Data availability
All data supporting the findings of this study are presented either in the paper, Supplementary Information or Source Data file. The PG profile dataset from the conditions screening of *V. cholerae* generated in this study is provided in the Supplementary Data 1 file and has been deposited in the Figshare repository, with https://doi.org/10.6084/m9.figshare.26807920. All strains are available upon request. Source data are provided with this paper.

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

## Acknowledgements
We thank all the members of the Cava lab for helpful discussions. Special thanks to Akbar Espaillat for providing the pET28b::*slt70Ec* and pBAD::*ldtE* plasmids, Nikola Zlatkov Kolev for providing the lambda phage, Alexander Harms for providing phages P1, P2, T4 and T5, and the Feldman lab for providing the Tse4 effector protein from *Acinetobacter baumannii*. We thank Michael Gilmore for carefully reading the manuscript. Research in the Cava lab is supported by The Swedish Research Council (VR), The Knut and Alice Wallenberg Foundation (KAW), The Laboratory of Molecular Infection Medicine Sweden (MIMS) and The Kempe Foundation. Work in the Dörr lab is supported by NIH R01GM130971.

## Author contributions
Conceptualization, L.A., S.B.H., and F.C.; Methodology, L.A., S.B.H., G.T., and A.I.W.; Investigation, L.A., S.B.H., G.T., and A.I.W.; Formal Analysis, L.A., and S.B.H.; Writing – Original Draft, L.A., and S.B.H; Writing – Review & Editing, L.A., S.B.H., G.T., A.I.W., T.D., and F.C.; Visualization, L.A., and S.B.H; Supervision, T.D., and F.C.; Resources, F.C.; Funding Acquisition, F.C.

## Funding

## Competing interests
The authors declare no competing interests.
