## [Peer Review File · Nature Communications]

REVIEWER COMMENTS

Reviewer #1 (Remarks to the Author):

The manuscript by Alvarez and co-workers reports a very interesting observation namely that the increase of LD type of crosslinks in the peptidoglycan (PG) results in a reduced ability of lytic transglycosylases (LTs) to hydrolyze PG and release the corresponding anhydromuropeptides. The authors discover the inverse correlation between the level of LD crosslinks and that of anhydromuropeptides by analyzing the PG composition of *V. cholerae* grown under more than 100 relevant physiological conditions. This inverse correlation seems not restricted to *V. cholerae* but found also in other Gram-negative species as shown by the increased level of anhydromuropeptides in bacteria treated with copper, a known inhibitor of LD transpeptidases among several other metallo-proteins. In vitro data also support the inverse correlation as shown by the reduced activity of lytic transglycosylases towards PG carrying increased degrees of LD crosslinks. Based on this evidence the authors suggest that LD crosslinks modify locally the PG structure so that the processivity of LTs is reduced. Finally, the authors show that the level of LD crosslinks impacts on the ability of *E. coli* cells to survive to phage attack or resist to poisoning by exogenous lytic transglycosylases produced by predatory or competing bacteria. Overall, these observations underlie the multiple roles LD crosslinked regions may fulfil ranging from protection to adaptation to different growth conditions thus giving a broader biological significance to this type of crosslinks.

The work is solid and the experiments well designed. Few comments are reported below.

General comment

In several sentences in the manuscript text the authors state “LD crosslinking downregulates LTs activity” (lines 117-118) or “LD crosslinkingnegatively regulates lytic transglycosylase activity” (line 126) or “LD crosslinksserves as negative regulators of LT enzyme activity” (lines 151-152). However, as also stated by the authors in the discussion, the data reported in the manuscript suggest that LD crosslinks restrict the access of LTs to their substrate but do not negatively regulate LTs activity. Perhaps the authors might rephrase the above sentences.

Specific comments

Supplementary Fig. 4

The authors use copper to inhibit the activity of *V. cholerae* LdtA in vivo and to assess that under LdtA inhibition conditions the level of anhydromuropeptides increases. 1mM (panel c) is indicated as the working CuSO₄ concentration to inhibit LdtA in growing *V. cholerae* cells in rich medium. However, 1 mM CuSO₄ seems to be toxic for cell growth as judged by the strong decrease in OD₆₀₀ shown in panel h. At which time (or growth phase) is the OD recorded and the cells sampled? Results shown in panel h do not show a clear inverse correlation between the level of anhydromuropeptides and LD crosslinks.

Also, the OD profile as a function of CuSO₄ concentration shown in panel h seems to be in contrast with the growth profile of *V. cholerae* shown in panel d (LB) where no growth inhibition is observed even at the maximum CuSO₄ concentration used. Can the authors explain these discrepancies?

CuSO₄ has a broader inhibitory effect on several metallo-proteins that goes beyond inhibition of LD transpeptidases implicated in the formation of LD crosslinks. The authors should measure the level of anhydromuropeptides in mutants missing LD transpeptidases and treated with CuSO₄. This would

exclude an indirect effect of copper on LTs.

Lanes 119-124 and supplementary Fig 8 panels b and c

It is not clear why the authors analyzed the level of LD crosslinks and anhydro-muropeptides in an *E. coli* mutant deleted for *ldtD* and *ldtF*; they should have used instead data relative to *ldtD* and *ldtE* double mutant whose PG analysis is reported in literature (Morè et al., 2019). Also, is the increase in anhydromuropeptides (0.11%, panel c) calculated for $\Delta ldtDF$ mutant from literature significant?

Lanes 143-144 and supplementary Fig. 9c

The authors expand the analysis of Vc LTs to *Slt70* from *E. coli* and claim “These findings were not specific to *V. cholerae*’s LTs as they were recapitulated using *E. coli*’s *Slt70*”. However, *Slt70* from *E. coli* seems not inhibited by the presence of wt level of LD-crosslinks in *V. cholerae* as *Slt70* from *V. cholerae* is (Figure 2 panel c). Perhaps inhibition of Ec *Slt70* by LD crosslinks should be assessed using PG purified from wild type *E. coli* and from the isogenic *ldt* mutant (producing 0% LD crosslinks).

Discussion

The data presented by the authors are nicely discussed and the model for LD-crosslinks mediated regulation of LTs activity fits very well with their findings. However, this section lacks a discussion of the conditions in which the inverse correlation between high level of LD-crosslinks and lower amounts of anhydrous was found. Why do cells regulate the production of LD-crosslinks and consequently the length of PG chains during growth in minimal medium? Could it be a specific mechanism related to growth under minimal medium?

Also, according to literature, is decrease in LD crosslink always accompanied by increase of anhydromuropeptides or vice versa? In a previous work by the same authors (Hernandez et al., 2014) the decrease in LD crosslinks following sodium deoxycholate treatment of *S. enterica* cells seems not accompanied by a significant increase in anhydromuropeptides. Can the authors comment?

Minor points

Supplementary figure 8 panel a

Data would be more readable with a table summarising the LD-crosslinks and Anhydro species found in the analyses.

Supplementary figure 8 panel b and c

Data for *M. abscessus* need to be revised, the authors consider a muropeptide containing DD (3-4) crosslinks (table 1 from Lavollay et al., 2011) that should not be included in the sum of LD crosslinks. Correct sum of 3-3 (LD) species should be: Rough exp. 29; Rough stat. 27; Smooth exp. 37; Smooth stat. 33.

Lane 370: thorough should be: through.

Reviewer #2 (Remarks to the Author):

In this manuscript, Alvarez et al report an interesting negative correlation between the activity of lytic transglycosylases (LTs), which cleave the glycan strands in the peptidoglycan (PG) sacculus, and the abundance of 3-3 crosslinks in the PG. They initially made the observation from a powerful high throughput assay for detecting changes in PG composition across different growth conditions. Here, decreased products of LT activity, i.e. anhydromuropeptides are observed in conditions that lead to increased 3-3 crosslinks in the PG in *V. cholerae*. The observation was validated using mutants deficient in 3-3 crosslinks or an inhibitor of 3-3 crosslinking enzyme Ldt. The inverse correlation was also found to be conserved in the PG composition of several Gram-negative bacterial species. Meanwhile, changes in LT product had no effect on 3-3 crosslinking, indicating a causal link between 3-3 crosslinks and ability of LTs to function. The in vivo observation is supported by biochemistry using several endolytic as well as exolytic LTs, which poorly cleave PG with high abundance of 3-3 crosslinks in vitro. This observation is specific to LTs and not other glycan hydrolases such as lysozyme. The authors have done extensive analysis which establishes that PG with high abundance of 3-3 crosslinks is a poor substrate for all LTs tested, including those encoded by phages.

Although, these observations are interesting, this study is preliminary and does not offer any mechanistic insights into the phenomenon.

A major drawback of the manuscript is that the authors claim that the activity of LTs is regulated by the 3-3 crosslinks. Point to be noted here is that the substrate specificity does NOT imply regulation. It requires evidence for physiologically relevant control mechanisms for homeostasis or to respond to environmental cues. Throughout the manuscript, the authors strongly refer to a regulatory mechanism instead of substrate specificity. Clearly more evidence is required to justify the statements made in the manuscript.

Questions/ comments and potential experiments to test the regulation hypothesis are listed below:

1. LTs having a higher preference for uncross-linked glycans is well documented (PMID: 23421439, PMID: 25480295, PMID: 35073258, PMID: 38422114), and supports their function in PG expansion, turnover, release of soluble fragments, mitigation of periplasmic crowding etc. Minor increase in LT activity with decreased 3-3 crosslinks is well in line with these observations, especially if 3-3 crosslinks are enriched in certain areas such as poles, their absence likely has an effect on total LT activity. The authors state that while 3-3 crosslinks are depleted, total crosslinking remains the same (line 144), however no such clear evidence is presented in their data. (on a related note, each of the references mentioned above are highly relevant to the present study and must be cited – one is from some of the authors themselves!)
2. In continuation, excess 3-3 crosslinks are lethal. As Ldts are able to form crosslinks in an existing mature PG sacculus, it is very likely that excess Ldt may form crosslinks using free tetrapeptides that are not oriented in the plane of the PG sacculus and therefore not available for crosslinking in the WT condition. One can imagine that such aberrant 3-3 crosslinks very likely change the overall architecture of PG, also making it a poorer substrate for LTs. Therefore, the decreased LT activity on the synthetic sacculi with 3-3 crosslink levels not normally found in cells (Fig. 2a, b) may not have any major physiological relevance.
3. The basis for decreased phage susceptibility due to LdtE overexpression is unclear. Increased 3-3 crosslinks are believed to increase the overall rigidity of the PG sacculus and may have an effect on phage entry. If the decreased susceptibility to phage infection is due to inhibition of phage LT, no difference would be expected in the susceptibility to phages that use other PG hydrolases such as

lysozyme, amidase etc for entry. Whether the mild resistance is specific to LT-encoding phages should be tested.

4. Have the authors observed any instance of 3-3 crosslinks or Ldt levels increasing in response to phage infection?

5. Although the activity of all LTs seems to be affected by 3-3 crosslinks in vitro, the contribution of few LTs may be more relevant to cell physiology than others. The glycan chain length determinant MltG contributes significantly to the native level of anhydro-muropeptides in the cell (PMID: 26507882 and Fig. 1e, S7a). Does an increase in 3-3 crosslinks lead to an increase in average glycan strand length in the cell similar to Δ mltG?

6. The basis for normalization of LT activity % is not clear. If LT activity on PG with 0% 3-3 crosslinks is considered 100% (like in fig 2 b,c), what is the LT activity in fig 2d relative to? In fig 9c, LT activity on WT PG is considered to be 100% for *V. cholerae*, while it's the opposite for *P. damsela*. How was the normalization done here?

7. Line 143 (These findings were not specific to *V. cholerae*'s LTs as they were recapitulated using *E. coli*'s Slt70) – Fig S9c is contradictory to the statement for *V. cholerae* PG.

8. Line 74-77: Would be appropriate to cite PMID: 38422114 for demonstrating a regulatory mechanism for a LT with a role in PG expansion.

9. Line 211: Processivity of the exolytic Slt being dependent on the crosslinking status has been clearly demonstrated in PMID: 25480295 and the reference should be cited.

10. Introduction is very general, rudimentary and does not describe the objectives of the study.

11. Several relevant citations are missing in the Introduction.

Reviewer #3 (Remarks to the Author):

This manuscript by Alvarez et al. uncovers a negative correlation between LD-crosslinking and the activity of lytic transglycosylase (LTs). First, the authors subjected *V. cholerae* to various environmental conditions and analyzed the changes in PG composition with a high-throughput HPLC approach. The amount of work is impressive. The strong negative correlation between LD-crosslinking and LT activity was reproduced when the authors treated the cells with copper and confirmed by deletion mutants. A series of carefully executed biochemical assays were conducted to support the initial observation and demonstrated that LD-crosslinking could serve as a protective mechanism by antagonizing the LTs from bacteria and phages. Understanding the mechanism that regulates cell wall synthesis is fundamental because it is a crucial process for maintaining the integrity of bacterial cells. PG synthesis is also an important target for antibiotics. Assessing bacterial PG composition under varied growth conditions can provide insights into the regulatory functions of synthetic and degradative enzymes involved in PG growth.

While the evidence supporting that LD crosslinked PG is not a preferred substrate of LTs is convincing, we are unsure if LD-crosslinking in PG negatively regulates LT activity. To qualify for a regulatory mechanism, one will need to show the cell can adjust LD-crosslinking in response to a signal. For example, figure 3 shows that overexpression of YnhG can block phage LTs. Yet, we don't see evidence showing the cell upregulates LD TPase when exposed to phages.

Another critical information missing is that some LTs are not quite active against DD crosslinked PG (PMID: 34036206). The same is true for similar enzymes, MpgA and MpgB, in *S. pneumoniae*, which are derivatives of LTs that become muramidases (PMID: 34475211). The new information provided by this work is thus in line with these findings.

That said, this study employs a comprehensive set of genetics and biochemical assays to establish the correlation between LD-crosslinking and LTs. The findings are well presented; only a few points need to be addressed. For instance, the reviewers found that the study's rationale was not clearly stated, and there is insufficient background information on why certain PG-degrading enzymes were selected. Additionally, discussing the differential sensitivity of certain LTs to LD-crosslinking with structural insights (e.g., AF models) would enhance the study's depth.

Specific points:

Line 64-79: The introduction abruptly ends. Other regulatory mechanisms that control PG hydrolases are not discussed. Is the study investigating the regulation of LTs (Line 78), the physiological function of LDTs (Line 63), or both? Perhaps a transition paragraph that states the objective of the study, a brief description of the methods, and the main conclusions of the study could help before moving on to the results section.

Line 108-110: Complementation of the Δ ldt mutant should be performed to show that the elevated level of anhydromuropeptide can be reverted to a comparable level to the parent strain.

Line 148-153: The authors argue that LD crosslinking 'serves as negative regulators of LT enzyme activity,' and 'this regulatory effect was also evident in vivo' because the amount of soluble anhydromuropeptides released is higher in the Δ ldt mutant. These statements are overstated because there is a lack of evidence indicating that LD crosslinking is an active regulatory mechanism. Thus far, the data presented support that LD crosslinked PG is not a preferred substrate for the LTs tested. To demonstrate this, perhaps the authors can consider using beta-lactams, which will block DD transpeptidases and increase LD transpeptidation (PMID: 25480295).

Line 165: The rationale of using lysozyme and mutanolysin were not clearly stated. A brief background of these enzymes could be mentioned in the introduction. Was it serving as a control? Explanation of why lysozyme retained full activity was provided but not for mutanolysin.

Line 169: 'LD crosslinks specifically downregulate the activity of LTs, but not lysozymes.' is confusing if not inaccurate. Downregulation usually refers to gene expression. Moreover, this is a generalized statement based on two lysozymes (egg white lysozyme and mutanolysin). How about MpgA and MpgB?

Line 546-558: Figure 3c-e. ynhG was not used in the entire text but appeared in the figure. It is confusing to use ldtE (in figure legend) and ynhG (in figures) interchangeably.

Minor concern:

Line 34: "Moreover, we demonstrate that this regulation controls the release of immunogenic PG fragments ..." needs references. It may be put in line 150.

Lines 67-68: I suggest merging the paragraphs since they both refer to the PG-degrading enzymes.

Line 68: To provide more information about autolysin. What are the other classes of autolysins? How are they different from lytic transglycosylases (LTs)?

Line 75: Please provide descriptions of "transenvelope nanomachine" as it is not a widely used term.

Line 81-82: I suggest rephrasing the heading since the paragraph primarily focuses on correlation between LD-crosslinking and LT activity based on measuring anhydromuropeptide released, but did not report the actual "glycan chain length" of PG.

Line 90: To reference Supplemental Figure 2 together with Figure 1b since it displayed the entire heatmap.

Line 98-99: A lower concentration of CuSO₄ was used in the MM condition despite the working concentration being determined to be 1 mM (Supplemental figure 4c). Is 1mM of Cu²⁺ toxic to the cell in MM condition?

Line 111: It would be nice to indicate the p-values for Supplemental Figure 1b (LD crosslinking) and 1c to support the sentence "Further, despite comparable morphology and growth, ...".

Line 178: Please rephrase "Together, these results demonstrate that increasing the degree of LD-crosslinking can repel an attack by predatory LTs of both bacterial and phage origin."

Line 203: "However, our results indicate that even low levels of LD-crosslinking can significantly inhibit LT activity". I think the authors are referring to Fig. 2c. It is interesting that Slt is inhibited more than other LTs. Perhaps I missed the point, but is it possible that it is due to Slt being an exo-LT?

Line 213-215: "Our data indicate that PG containing DD-crosslinks is degraded more efficiently than PG containing LD-crosslinks." Please indicate the figure the authors is referring to.

Line 219-222: References that indicate *Agrobacterium tumefaciens*, *Mycobacterium tuberculosis*, or *P. damsela* PG having naturally high LD-crosslinked cell walls should be provided.

Line 242 and 254: The authors proposed that the LD crosslink could play a role in antibiotic resistance. However, in the high-throughput screen (Figure 1), the LD crosslink is not more pronounced under the

antibiotic stress tested. Is there any reason for this observation?

Line 283: Indicate the abbreviation MCS. I believe it is “Multiple cloning site”

Line 326: To correct the Parenthesis. Perhaps it is referring to “(100 µg/ml in water) overnight at 37°C.”?

Line 533: To define what ND is in the figure legend. i.e. ND: Not detected.

Line 541-543: Figure 2d. How is the relative LT activity level calculated?

Line 554-556: Figure 3d. What does '**' indicate?

Supplemental figure 10: Wrong indication of panel c in the figure legend.

Supplemental Figure 6c. Please correct the spacing issue. 'Δldt mutant cultures grown “overnight” in LB ...'

Reviewer #4 (Remarks to the Author):

Reviewer #5 (Remarks to the Author):

Reviewer #6 (Remarks to the Author):

Response to reviewers

Reviewer #1 (Remarks to the Author):

The manuscript by Alvarez and co-workers reports a very interesting observation namely that the increase of LD type of crosslinks in the peptidoglycan (PG) results in a reduced ability of lytic transglycosylases (LTs) to hydrolyze PG and release the corresponding anhydromuropeptides. The authors discover the inverse correlation between the level of LD crosslinks and that of anhydromuropeptides by analyzing the PG composition of *V. cholerae* grown under more than 100 relevant physiological conditions. This inverse correlation seems not restricted to *V. cholerae* but found also in other Gram-negative species as shown by the increased level of anhydromuropeptides in bacteria treated with copper, a known inhibitor of LD transpeptidases among several other metallo-proteins. In vitro data also support the inverse correlation as shown by the reduced activity of lytic transglycosylases towards PG carrying increased degrees of LD crosslinks. Based on this evidence the authors suggest that LD crosslinks modify locally the PG structure so that the processivity of LTs is reduced. Finally, the authors show that the level of LD crosslinks impacts on the ability of *E. coli* cells to survive to phage attack or resist to poisoning by exogenous lytic transglycosylases produced by predatory or competing bacteria. Overall, these observations underlie the multiple roles LD crosslinked regions may fulfil ranging from protection to adaptation to different growth conditions thus giving a broader biological significance to this type of crosslinks.

The work is solid and the experiments well designed. Few comments are reported below.

We thank the reviewer for their detailed assessment of our manuscript and their enthusiastic comments.

General comment

In several sentences in the manuscript text the authors state “LD crosslinking downregulates LTs activity” (lines 117-118) or “LD crosslinkingnegatively regulates lytic transglycosylase activity” (line 126) or “LD crosslinksserves as negative regulators of LT enzyme activity” (lines 151-152). However, as also stated by the authors in the discussion, the data reported in the manuscript suggest that LD crosslinks restrict the access of LTs to their substrate but do not negatively regulate LTs activity. Perhaps the authors might rephrase the above sentences.

We agree with the reviewer that LD-crosslinks likely restrict the access of the LTs to their substrate. Hence, we have carefully revised the manuscript and rephrased those sentences (line numbers 158-159, 168, 195-197).

Specific comments

Supplementary Fig. 4

The authors use copper to inhibit the activity of *V. cholerae* LdtA in vivo and to assess that under LdtA inhibition conditions the level of anhydromuropeptides increases. 1mM (panel c) is indicated as the working CuSO_4 concentration to inhibit LdtA in growing *V. cholerae* cells in rich medium. However, 1 mM CuSO_4 seems to be toxic for cell growth as judged by the strong decrease in OD600 shown in panel h. At which time (or growth phase) is the OD recorded and the cells sampled? Results shown in panel h do not show a clear inverse correlation between the level of anhydromuropeptides and LD crosslinks.

Also, the OD profile as a function of CuSO_4 concentration shown in panel h seems to be in contrast with the growth profile of *V. cholerae* shown in panel d (LB) where no growth inhibition is observed even at the maximum CuSO_4 concentration used. Can the authors explain these discrepancies?

We thank the reviewer for pointing this out. The decrease in OD₆₀₀ originally shown in Fig. S4h corresponded to a \$\text{CuSO}_4\$ concentration of 1.25 mM. We have observed a drop in OD₆₀₀ at concentrations higher than 1 mM in LB, in agreement with the growth curves shown in Fig. S4d, where the maximum \$\text{CuSO}_4\$ concentration tested was 2 mM and was the only one producing a defect in growth.

To improve the data presentation and more effectively show the inverse correlation between LD-crosslinking and the level of anhydromuropeptides, we have prepared new PG samples at different CuSO_4 concentrations (ranging from 0.1 to 1 mM) and measured the OD_{600} . These results, shown in the new Fig. S4h, are consistent with those presented in S4d. We have chosen a log scale for the y-axis for better comparison with the data presented in panel S4d. The y-axis of the plot in Fig. 1d has been changed accordingly.

Furthermore, we have also repeated the growth curves with different CuSO_4 concentrations, and the results recapitulate those shown before (Fig. R1):

Fig. R1: Growth curves of *V. cholerae* WT grown in LB with different concentrations of CuSO_4 . Error bars represent standard deviation.

CuSO_4 has a broader inhibitory effect on several metallo-proteins that goes beyond inhibition of LD transpeptidases implicated in the formation of LD crosslinks. The authors should measure the level of anhydromuropeptides in mutants missing LD transpeptidases and treated with CuSO_4 . This would exclude an indirect effect of copper on LTs.

The reviewer raises a valid point. To address this concern, we have analyzed the PG of the Δldt mutant with and without CuSO_4 both in LB and MM media. Our results show that there are no significant differences in the levels of anhydromuropeptides upon treatment with CuSO_4 (Fig. R2). We have added this important control in the new Fig. S6c.

Fig. R2: Relative amount of anhydromuropeptides in the PG of the *V. cholerae* Δldt mutant grown in LB and MM supplemented or not with CuSO_4 . Error bars represent standard deviation.

Lanes 119-124 and supplementary Fig 8 panels b and c

It is not clear why the authors analyzed the level of LD crosslinks and anhydro-muropeptides in an *E. coli* mutant deleted for ldtD and ldtF ; they should have used instead data relative to ldtD and ldtE double mutant whose PG analysis is reported in literature (Morè et al., 2019). Also, is the increase in anhydromuropeptides (0.11%, panel c) calculated for ΔldtDF mutant from literature significant?

We understand the reviewer's point as ldtD and ldtE are *E. coli*'s DAP-DAP crosslinking LDTs. However, in our review of the literature, we focused on samples with a decrease in LD-crosslink and the $\Delta\text{ldtD}\Delta\text{ldtF}$ mutant reported by Morè et al. satisfied this criterium while the $\Delta\text{ldtD}\Delta\text{ldtE}$ mutant did not (Fig. R3).

Fig. R3: Changes in LD-crosslink and anhydromuropeptide content reported by Morè et al. (2019).

We cannot conclusively determine the importance of the increased anhydromuropeptides in the $\Delta ldtD\Delta ldtF$ mutant because the original publication only reported a single set of results, making statistical analysis impossible. However, we notice a consistent pattern supported by our own findings in *E. coli*, with and without $CuSO_4$ treatment (Fig. 2f). This pattern is also confirmed by the data from the *E. coli* BW25113 Δldt mutant presented in the updated Figure S9ab.

Lanes 143-144 and supplementary Fig. 9c

The authors expand the analysis of Vc LTs to Slt70 from *E. coli* and claim “These findings were not specific to *V. cholerae*’s LTs as they were recapitulated using *E. coli*’s Slt70”. However, Slt70 from *E. coli* seems not inhibited by the presence of wt level of LD-crosslinks in *V. cholerae* as Slt70 from *V. cholerae* is (Figure 2 panel c). Perhaps inhibition of Ec Slt70 by LD crosslinks should be assessed using PG purified from wild type *E. coli* and from the isogenic *ldt* mutant (producing 0% LD crosslinks).

We have complemented Fig. S9 by incorporating the data from the in vitro assay of Slt70_{Ec} on sacculi derived from *E. coli* BW25113 WT, which exhibits 12% LD-crosslinks, and *E. coli* BW25113 Δldt mutant (PMID: 28974693), which is devoid of LD-crosslinks. The updated findings substantiate that Slt70_{Ec}’s activity is diminished on sacculi with LD-crosslinks (new Fig. S9c).

In addition, to further test our hypothesis, we performed additional in vitro assays of Slt70_{Ec} on sacculi showing a gradation of LD-crosslinking levels from 0% up to 17.5% (presented in the new Fig. S9d). These supplementary experiments provide robust support for our hypothesis, illustrating a definitive trend in the impact of LD-crosslinking on Slt70_{Ec}’s enzymatic activity.

Discussion

The data presented by the authors are nicely discussed and the model for LD-crosslinks mediated regulation of LTs activity fits very well with their findings. However, this section lacks a discussion of the conditions in which the inverse correlation between high level of LD-crosslinks and lower amounts of anhydrous was found. Why do cells regulate the production of LD-crosslinks and consequently the length of PG chains during growth in minimal medium? Could it be a specific mechanism related to growth under minimal medium?

Also, according to literature, is decrease in LD crosslink always accompanied by increase of anhydromuropeptides or vice versa? In a previous work by the same authors (Hernandez et al., 2014) the decrease in LD crosslinks following sodium deoxycholate treatment of *S. enterica* cells seems not accompanied by a significant increase in anhydromuropeptides. Can the authors comment?

We propose that LD-crosslinks increase under specific stress conditions to limit LTs from acting on PG. In *Vibrio cholerae*, the only LDT that forms DAP-DAP crosslinks is LdtA, regulated by RpoS. This is evident from the inverse relationship between LD-crosslinks and anhydromuropeptides in environments controlled by RpoS, such as minimal media and high pH, but also in copper-containing media which inhibits LDT activity. Yet, there are scenarios where this inverse relationship does not hold, indicating that LTs may be regulated independently. As the reviewer pointed out, one such example is the effect of bile on the PG. Our previous studies showed that bile causes a reduction of the LD-levels in the PG of *Salmonella* (PMID: 24762004). Our current screening reveals a similar pattern in *V. cholerae*, but without an increase in anhydromuropeptide levels, suggesting that bile may also, directly or indirectly,

affect the function or production of LTs in this bacterium. We have updated the discussion to address the regulation of LDTs under various environmental conditions and included observations on the impact of bile on anhydromuropeptide levels in *Salmonella* (lines 306-326).

Minor points

Supplementary figure 8 panel a

Data would be more readable with a table summarising the LD-crosslinks and Anhydro species found in the analyses.

We understand the reviewer's concern. However, the quantifications of LD-crosslink and anhydromuropeptide level corresponding to the profiles shown in Fig. S8a are already shown in Fig. 1f, and the values are provided in the Source Data file.

Supplementary figure 8 panel b and c

Data for *M. abscessus* need to be revised, the authors consider a muropeptide containing DD (3-4) crosslinks (table 1 from Lavollay et al., 2011) that should not be included in the sum of LD crosslinks. Correct sum of 3-3 (LD) species should be: Rough exp. 29; Rough stat. 27; Smooth exp. 37; Smooth stat. 33.

We thank the reviewer for pointing this out. We have corrected the data in the figure.

Lane 370: thorough should be: through.

Thanks, we have corrected the typo (line 454).

Reviewer #2 (Remarks to the Author):

In this manuscript, Alvarez et al report an interesting negative correlation between the activity of lytic transglycosylases (LTs), which cleave the glycan strands in the peptidoglycan (PG) sacculus, and the abundance of 3-3 crosslinks in the PG. They initially made the observation from a powerful high throughput assay for detecting changes in PG composition across different growth conditions. Here, decreased products of LT activity, i.e. anhydromuropeptides are observed in conditions that lead to increased 3-3 crosslinks in the PG in *V. cholerae*. The observation was validated using mutants deficient in 3-3 crosslinks or an inhibitor of 3-3 crosslinking enzyme Ldt. The inverse correlation was also found to be conserved in the PG composition of several Gram-negative bacterial species. Meanwhile, changes in LT product had no effect on 3-3 crosslinking, indicating a causal link between 3-3 crosslinks and ability of LTs to function. The in vivo observation is supported by biochemistry using several endolytic as well as exolytic LTs, which poorly cleave PG with high abundance of 3-3 crosslinks in vitro. This observation is specific to LTs and not other glycan hydrolases such as lysozyme. The authors have done extensive analysis which establishes that PG with high abundance of 3-3 crosslinks is a poor substrate for all LTs tested, including those encoded by phages.

Although, these observations are interesting, this study is preliminary and does not offer any mechanistic insights into the phenomenon.

A major drawback of the manuscript is that the authors claim that the activity of LTs is regulated by the 3-3 crosslinks. Point to be noted here is that the substrate specificity does NOT imply regulation. It requires evidence for physiologically relevant control mechanisms for homeostasis or to respond to environmental cues. Throughout the manuscript, the authors strongly refer to a regulatory mechanism instead of substrate specificity. Clearly more evidence is required to justify the statements made in the manuscript.

We thank the reviewer for their detailed assessment of our manuscript. In the revised version of our manuscript, we have tried to clarify every point raised. We have also toned down the use of “regulation” and made clearer that LD-crosslinks interfere with LTs’ ability to access their substrate, resulting in reduced activity.

Questions/ comments and potential experiments to test the regulation hypothesis are listed below:

1. LTs having a higher preference for uncross-linked glycans is well documented (PMID: 23421439, PMID: 25480295, PMID: 35073258, PMID: 38422114), and supports their function in PG expansion, turnover, release of soluble fragments, mitigation of periplasmic crowding etc. Minor increase in LT activity with decreased 3-3 crosslinks is well in line with these observations, especially if 3-3 crosslinks are enriched in certain areas such as poles, their absence likely has an effect on total LT activity. The authors state that while 3-3 crosslinks are depleted, total crosslinking remains the same (line 144), however no such clear evidence is presented in their data. (on a related note, each of the references mentioned above are highly relevant to the present study and must be cited – one is from some of the authors themselves!)

We thank the reviewer for their insightful comments. The suggested references have now been duly incorporated into the manuscript (line 266).

While previous studies have indicated a preference of LTs for uncrosslinked PG, our research brings to light a more pronounced inhibitory effect of LD-crosslinks as compared to DD-crosslinks on LT activity. This was shown at different levels. First, our PG chemical profiling screen did not find any significant inverse relationship between high DD-crosslinking and anhydromuropeptide levels (Fig. S3 and Fig. R4). Instead, we observed that lower DD-crosslinking correlates with increased LD-crosslinking, maintaining overall crosslinking balance, and these conditions also correspond with reduced anhydromuropeptide levels. Moreover, in vitro assays comparing WT and Δldt sacculi further confirm LTs demonstrate a clear predilection for PG with fewer LD-crosslinks. This specificity is evident even when comparing substrates with similar total crosslink percentages but different LD/DD ratios, as shown in the new Figure S9b. These quantifications, included in our revised manuscript, provide concrete evidence of this selective degradation by LTs, emphasizing the critical role of crosslink composition in LT activity.

Fig. R4: Selected plots from Fig. S3 showing the relationship between DD- and LD-crosslinking with anhydromuropeptide content and total crosslinking.

2. In continuation, excess 3-3 crosslinks are lethal. As Ldts are able to form crosslinks in an existing mature PG sacculus, it is very likely that excess Ldt may form crosslinks using free tetrapeptides that are not oriented in the plane of the PG sacculus and therefore not available for crosslinking in the WT condition. One can imagine that such aberrant 3-3 crosslinks very likely change the overall architecture of PG, also making it a poorer substrate for LTs. Therefore, the decreased LT activity on the synthetic sacculi with 3-3 crosslink levels not normally found in cells (Fig. 2a, b) may not have any major physiological relevance.

We respectfully disagree with the reviewer that elevated 3-3 crosslinks is fatal. It is well-documented that numerous bacteria thrive with high levels of LD-crosslinking. For instance, *Agrobacterium tumefaciens* (PMID: 22307633), *Mycobacterium tuberculosis* (PMID: 18408028), and *Photobacterium damselae* (PMID: 33536321) all exhibit substantial LD-crosslinking. Additionally, *Vibrio cholerae* significantly increases its LD-crosslink percentage in response to environmental conditions, from 1.5% in LB to 25% in MM. Furthermore, the induced expression of LdtA leads to an elevation of LD-crosslinking to 16% in LB and a striking 48% in MM, observable in both the WT strain and the Δldt mutant (as shown in Fig. R5B). Our study's findings align with these patterns, indicating that a rise in LD-crosslinking correlates with reduced anhydromuropeptide levels (Fig. R5C). Importantly, these substantial alterations in PG composition do not impede growth, as evidenced by the consistent OD_{600} measurements across all cultures (illustrated in Fig. R5A).

Fig. R5: Analysis of growth (A), relative LD-crosslinking (B) and anhydromuropeptide levels (C) in *V. cholerae* WT and Δldt mutant and complemented strains grown in LB and MM.

We concur with the reviewer that a naturally modified sacculus differs from the artificially altered one we have created using LdtA. However, both types of substrates are instrumental in our study. The use of synthetically modified sacculi narrows the variability to just the LD-crosslink quantity, thereby minimizing the impact of other potential variances in PG composition on LT activity. It is also important to note that the maximum LD-crosslink percentage achieved in the LdtA-modified substrate is 17.5%, which is actually below the natural LD-crosslink levels observed in *V. cholerae* cultivated in nutrient-

deficient media, or in other bacterial species such as *P. damselae*. This comparison underscores the relevance and applicability of our synthetic PG in studying the effect of LD-crosslinking on LT activity.

3. The basis for decreased phage susceptibility due to LdtE overexpression is unclear. Increased 3-3 crosslinks are believed to increase the overall rigidity of the PG sacculus and may have an effect on phage entry. If the decreased susceptibility to phage infection is due to inhibition of phage LT, no difference would be expected in the susceptibility to phages that use other PG hydrolases such as lysozyme, amidase etc for entry. Whether the mild resistance is specific to LT-encoding phages should be tested.

We appreciate the reviewer's insightful suggestion, which prompted us to utilize the Basel *E. coli*-phage collection (PMID: 34784345) for a comprehensive analysis. Initially, we examined the genomes of all phages within the collection, pinpointing those encoding endolysins. These were then categorized based on homology and functional domains. We finally selected a diverse array of phages, each representing a distinct endolysin type, and proceeded with our experiments. We are grateful to Alexander Harms for providing us with phages P2, P1, T4, and T5, which encode LT-like, lysozyme-like endolysins, and an endopeptidase, respectively.

Mirroring the methodology applied in the lambda phage experiments in Fig. 3, we conducted infection assays with these additional phages. The findings, detailed in Fig. 3e and Supplementary Fig. S12, underscored a heightened resistance to LT-encoding phages, a pattern not observed with phages encoding other peptidoglycan hydrolytic enzymes like lysozymes or endopeptidases.

We thank the reviewer for this constructive challenge, which has significantly enriched our study and confirmed key aspects of our hypothesis.

4. Have the authors observed any instance of 3-3 crosslinks or Ldt levels increasing in response to phage infection?

This is an interesting and technically challenging question. We performed lambda phage infections and took 2 time points for analysis: the initial moment when the optical density begins to decrease due to phage activity, and a subsequent point at 24 hours. To ensure a robust comparison, we included controls at these exact time points for uninfected *E. coli* maintained under identical culture conditions (Fig. R6A). Our findings reveal that LD-crosslinking does not increase as a result of phage infection (Fig. R6B). These results are further supported by an RNAseq analysis of the transcriptional response of *E. coli* to lambda phage infection showed that none of the LDTs was differentially expressed (PMID: 37653008). We have included Figure R6 in the revised manuscript as Fig. S11de.

Altogether, this suggests that while PG containing LD-crosslinks exhibits greater resilience to LT-mediated phage attacks, such resistance is not a direct response to the phage itself but is influenced by environmental factors, such as nutrient availability or outer membrane damage, as supported by previous research (PMID: 21792174, PMID: 30723128). However, given the variation in the number and regulatory mechanisms of LDTs across different species (PMID: 34109739, doi: <https://doi.org/10.1101/2024.06.21.600065>), it's plausible that external LT attacks, such as those from specific phages, may provoke a defensive LD-crosslinking reaction in some species. We have explored this possibility in the discussion section (lines 299-317).

Fig. R6: Variation of LD-crosslinking during phage infection. A) *E. coli* JM109 was infected with lambda phage and PG samples were collected and analyzed at the indicated time points. B) Variation in LD-crosslinking is not significant (paired t-test, p-value = 0.4914).

5. Although the activity of all LTs seems to be affected by 3-3 crosslinks in vitro, the contribution of few LTs may be more relevant to cell physiology than others. The glycan chain length determinant MltG contributes significantly to the native level of anhydro-muropeptides in the cell (PMID: 26507882 and Fig. 1e, S7a). Does an increase in 3-3 crosslinks lead to an increase in average glycan strand length in the cell similar to Δ mltG?

Indeed, an increase in LD-crosslinks leads to a decrease in anhydromuropeptides (i.e., increase the average glycan strand length). To further support this, we have analyzed both *V. cholerae* WT and Δ ldt complemented strains. These results are presented in Fig. 1e and Fig. S6a.

6. The basis for normalization of LT activity % is not clear. If LT activity on PG with 0% 3-3 crosslinks is considered 100% (like in fig 2 b,c), what is the LT activity in fig 2d relative to? In fig 9c, LT activity on WT PG is considered to be 100% for *V. cholerae*, while it's the opposite for *P. damsela*e. How was the normalization done here?

Thank you for bringing this to our attention. To clarify, we have standardized the normalization of the LT activity across all figures for consistency. Initially, LT activity in Fig. 2d was benchmarked against the 100% activity observed on *V. cholerae* Δ ldt sacculi, which have 0% LD-crosslinks. This same reference point was applied to the in vitro assays for *P. damsela*e WT and Δ ldt sacculi. In the original Fig. S9c, the baseline for 100% LT activity was set using *P. damsela*e Δ ldt sacculi due to its higher activity levels.

To improve understanding and readability, we have recalibrated the LT activity in all relevant figures to be relative to the sacculi substrate with 0% LD-crosslinks. This adjustment ensures a uniform reference point across our data presentation. Consequently, we have updated Fig. 2cd, S9cd, and S10bc to reflect this change. We believe this provides a clearer and more coherent interpretation of our results.

7. Line 143 (These findings were not specific to *V. cholerae*'s LTs as they were recapitulated using *E. coli*'s Slit70) – Fig S9c is contradictory to the statement for *V. cholerae* PG.

We have complemented Fig. S9 by incorporating the data from the in vitro assay of Slit70_{Ec} on sacculi derived from *E. coli* BW25113 WT, which exhibits 12% LD-crosslinks, and *E. coli* BW25113 Δ ldt mutant (PMID: 28974693), which is devoid of LD-crosslinks. The updated findings substantiate that Slit70_{Ec}'s activity is diminished on sacculi with LD-crosslinks (new Fig. S9c).

In addition, to reinforce our conclusions, we performed additional in vitro assays of Slit70_{Ec} on sacculi showing a gradation of LD-crosslinking levels from 0% up to 17.5% (presented in the new Fig. S9d). These supplementary experiments provide robust support for our hypothesis, illustrating a definitive trend in the impact of LD-crosslinking on Slit70_{Ec}'s enzymatic activity, similarly to Slit70_{Vc}.

8. Line 74-77: Would be appropriate to cite PMID: 38422114 for demonstrating a regulatory mechanism for a LT with a role in PG expansion.

We thank the reviewer for the suggestion and have added the reference (line 90).

9. Line 211: Processivity of the exolytic Slit being dependent on the crosslinking status has been clearly demonstrated in PMID: 25480295 and the reference should be cited.

We have cited the reference (line 266).

10. Introduction is very general, rudimentary and does not describe the objectives of the study.

We have revised the introduction and included a final paragraph summarizing the main results of the study.

11. Several relevant citations are missing in the Introduction.

We have revised the introduction and included relevant references.

Reviewer #3 (Remarks to the Author):

This manuscript by Alvarez et al. uncovers a negative correlation between LD-crosslinking and the activity of lytic transglycosylase (LTs). First, the authors subjected *V. cholerae* to various environmental conditions and analyzed the changes in PG composition with a high-throughput HPLC approach. The amount of work is impressive. The strong negative correlation between LD-crosslinking and LT activity was reproduced when the authors treated the cells with copper and confirmed by deletion mutants. A series of carefully executed biochemical assays were conducted to support the initial observation and demonstrated that LD-crosslinking could serve as a protective mechanism by antagonizing the LTs from bacteria and phages. Understanding the mechanism that regulates cell wall synthesis is fundamental because it is a crucial process for maintaining the integrity of bacterial cells. PG synthesis is also an important target for antibiotics. Assessing bacterial PG composition under varied growth conditions can provide insights into the regulatory functions of synthetic and degradative enzymes involved in PG growth.

While the evidence supporting that LD crosslinked PG is not a preferred substrate of LTs is convincing, we are unsure if LD-crosslinking in PG negative regulates LT activity. To qualify for a regulatory mechanism, one will need to show the cell can adjust LD-crosslinking in response to a signal. For example, figure 3 shows that overexpression of YnhG can block phage LTs. Yet, we don't see evidence showing the cell upregulates LD TPase when exposed to phages.

Another critical information missing is that some LTs are not quite active against DD crosslinked PG (PMID: 34036206). The same is true for similar enzymes, MpgA and MpgB, in *S. pneumoniae*, which are derivatives of LTs that become muramidases (PMID: 34475211). The new information provided by this work is thus in line with these findings.

That said, this study employs a comprehensive set of genetics and biochemical assays to establish the correlation between LD-crosslinking and LTs. The findings are well presented; only a few points need to be addressed. For instance, the reviewers found that the study's rationale was not clearly stated, and there is insufficient background information on why certain PG-degrading enzymes were selected. Additionally, discussing the differential sensitivity of certain LTs to LD-crosslinking with structural insights (e.g., AF models) would enhance the study's depth.

We thank the reviewer for the careful reading of our manuscript and their positive comments. In the revised manuscript, we have included new experiments and further clarifications to address all concerns. Briefly:

Inducibility of LD-crosslinking by LT-encoding phages. We performed lambda phage infections and took 2 time points for analysis. Our findings (included now as Suppl. Figure 11de of the revised manuscript) show that LD-crosslinking does not increase as a result of lambda phage infection (Fig. R6B), which is in agreement with an RNAseq analysis of the transcriptional response of *E. coli* to lambda phage infection showing that none of the LDTs was differentially expressed (PMID: 37653008). This indicates that while PG containing LD-crosslinks exhibits greater resilience to LT-mediated phage attacks, such resistance is not a direct response to the phage itself but is influenced by environmental factors, such as nutrient availability or outer membrane damage, as supported by previous research (PMID: 21792174, PMID: 30723128). However, given the variation in the number and regulatory mechanisms of LDTs across different species, it is still plausible that external LT attacks, such as those from specific phages, may provoke a defensive LD-crosslinking reaction in some species. We have also explored this possibility in the discussion section (lines 299-317).

Effect of crosslinking on LT activity. We also agree with the reviewer that previous studies have indicated a preference of LTs for uncrosslinked PG. We have added the corresponding references to the text, line 266. However, our research brings to light a more pronounced inhibitory effect of LD-crosslinks as compared to DD-crosslinks on LT activity. This was shown at different levels. First, our PG chemical profiling screen did not find any significant inverse relationship between high DD-crosslinking and anhydromuropeptide levels. Instead, we observed that lower DD-crosslinking correlates with increased LD-crosslinking, maintaining overall crosslinking balance, and these

conditions also correspond with reduced anhydromuropeptide levels (Fig. S3, Fig. R4). Moreover, in vitro assays comparing WT and Δ/dt sacculi further confirm LTs demonstrate a clear predilection for PG with fewer LD-crosslinks. This specificity is evident even when comparing substrates with similar total crosslink percentages but different LD/DD ratios, as shown in the new Fig. S9b. These quantifications, included in our revised manuscript, provide concrete evidence of this selective degradation by LTs, emphasizing the critical role of crosslink composition in LT activity.

Finally, we have revised the introduction and included a final paragraph summarizing the main results of the study.

We appreciate the reviewer's suggestion to incorporate AlphaFold predictions. Although we recognize the importance of comparing the sequence and structure of different LTs and their interactions with LD-crosslinks, we prefer to defer this analysis to future research to avoid speculating about the structural determinants of LT sensitivity to LD-crosslinks without supporting data.

Specific points:

Line 64-79: The introduction abruptly ends. Other regulatory mechanisms that control PG hydrolases are not discussed. Is the study investigating the regulation of LTs (Line 78), the physiological function of LDTs (Line 63), or both? Perhaps a transition paragraph that states the objective of the study, a brief description of the methods, and the main conclusions of the study could help before moving on to the results section.

As suggested by the reviewer, we have revised the introduction and included a final paragraph that summarizes the objectives of the study, methods used and conclusions.

Line 108-110: Complementation of the Δ/dt mutant should be performed to show that the elevated level of anhydromuropeptide can be reverted to a comparable level to the parent strain.

As requested by the reviewer, we have included the results of the complemented mutant in Fig. 1e and Fig. S6a.

Line 148-153: The authors argue that LD crosslinking 'serves as negative regulators of LT enzyme activity,' and 'this regulatory effect was also evident in vivo' because the amount of soluble anhydromuropeptides released is higher in the Δ/dt mutant. These statements are overstated because there is a lack of evidence indicating that LD crosslinking is an active regulatory mechanism. Thus far, the data presented support that LD crosslinked PG is not a preferred substrate for the LTs tested. To demonstrate this, perhaps the authors can consider using beta-lactams, which will block DD transpeptidases and increase LD transpeptidation (PMID: 25480295).

The reviewer has raised a very valid point. As mentioned above, we have included a new supplementary Fig. S11de showing that *E. coli* does not increase LD-crosslinking during lambda infection and discussed these results in lines 302-308. We have also toned down the text and changed the statements referring "regulation" for "interference with activity" or similar.

We have not used beta-lactams to block DD-transpeptidases and increase LD-transpeptidation because in our conditions screening there is no significant increase in LD-crosslinks upon treatment with diverse beta-lactams (Fig. S2).

Line 165: The rationale of using lysozyme and mutanolysin were not clearly stated. A brief background of these enzymes could be mentioned in the introduction. Was it serving as a control? Explanation of why lysozyme retained full activity was provided but not for mutanolysin.

We tested lysozyme and mutanolysin as examples of other PG-degrading enzymes acting on the same β -1,4-glycosidic bond between NAM and NAG as LTs do. Both mutanolysin and chicken egg lysozyme remained similarly unaffected by increasing levels of LD-crosslinks and, since mutanolysin is a type of lysozyme from *Streptomyces*, we did not include a separate explanation for the mutanolysin behaviour. We have now clarified this in the text (lines 79-86, 211-216).

We have provided a reason why we use lysozyme and mutanolysin in the text (lines 209-211).

Line 169: 'LD crosslinks specifically downregulate the activity of LTs, but not lysozymes.' is confusing if not inaccurate. Downregulation usually refers to gene expression. Moreover, this is a generalized statement based on two lysozymes (egg white lysozyme and mutanolysin). How about MpgA and MpgB?

As commented above, we have toned down the references to "regulation" regarding the activity and referred to interference by LD-crosslinks on the activity instead throughout the text.

We thank also the reviewer for their insightful inquiry into the effects of LD-crosslinks on the activity of lysozymes such as MpgA and MpgB. These enzymes were predicted to have a LT-fold based on their homology to *E. coli*'s MltG. However, these proteins have lysozyme activity on the lipid-linked nascent peptidoglycan. MltG is a true LT, but the D245N mutation has been also reported to render lysozyme activity (PMID: 34475211).

To address this question, we requested Suzanne Walker the plasmids for purification of MpgA and MpgB from *Streptococcus pneumoniae* and MltG and MltG D245N from *E. coli* (Fig. R7A). We also mutated the same key residue in *V. cholerae*'s MltG protein and purified the corresponding MltG_{Vc} version.

We performed in vitro assays on *V. cholerae* sacculi to assess the type of PG degrading activity and surprisingly found that, under our experimental conditions, we only detected LT activity (only anhydromuropeptides were solubilized) by all these enzymes (Fig. R7B). Remarkably, we observed exclusively LT activity, with only anhydromuropeptides being solubilized (Fig. R7B). Liberation of M4 resulting from lysozyme activity was only detected in the muramidase treated sample. We then tested the impact of LD-crosslinks on the activity of the enzymes. Further experiments assessed the influence of LD-crosslinks on enzymatic activity. Consistent with previous findings for *V. cholerae*'s native LTs, we found that LT activity was substantially reduced in substrates with a higher concentration of LD-crosslinks (Fig. R7C).

It is important to note that our assay methodology diverges significantly from that used in the study referenced by PMID: 34475211, which employed lipid II and a glycosyltransferase to generate a substrate akin to nascent PG.

Although the primary goal of our assay was to investigate the impact of LD-crosslinking on other lysozymes, these enzymes did not exhibit such activity on mature sacculi, leading us to exclude these assays from our manuscript. Nevertheless, we have incorporated new phage infection experiments in Fig. 3, including phages P1 and T4 that encode endolysins with lysozyme activity. These experiments corroborate our in vitro findings: phages encoding LTs (and not those encoding lysozymes or other non-LT autolysins) are influenced by the levels of LD-crosslinking. These findings further substantiate our conclusion that LT activity, as opposed to lysozyme activity, is modulated by the presence of LD-crosslinked PG.

Fig. R7: In vitro activity of other lysozymes and lytic transglycosylases. A. Purified proteins. Predicted molecular weights are indicated underneath and corresponding bands are indicated with arrows when needed. B. Representative chromatograms showing the released muropeptides after digestion of *V. cholerae* sacculi with muramidase (Mur.), MpgA and MpgB from *S. pneumoniae*, MltG and MltG D245N mutant from *E. coli*, and MltG and MltG D245N from *V. cholerae*. Plots show the BPI (base peak intensity) traces after MS-MS/MS analysis. C. LT activity on substrate with indicated LD-crosslinking levels. Total area of the peaks of released muropeptides (in arbitrary units, AU) is shown.

Line 546-558: Figure 3c-e. ynhG was not used in the entire text but appeared in the figure. It is confusing to use ldtE (in figure legend) and ynhG (in figures) interchangeably.

The reviewer is correct, and we have fixed the label in the figure.

Minor concern:

Line 34: "Moreover, we demonstrate that this regulation controls the release of immunogenic PG fragments ..." needs references. It may be put in line 150.

The effect of LD-crosslinking on the release of PG fragments is shown in Fig. S9e and discussed in lines 285-292. In those lines, we refer to the review by Irazoki et al (2019) (PMID: 30984120), where the role of released muropeptides in different processes is reviewed.

Lines 67-68: I suggest merging the paragraphs since they both refer to the PG-degrading enzymes.

We have revised and edited the introduction.

Line 68: To provide more information about autolysin. What are the other classes of autolysins? How are they different from lytic transglycosylases (LTs)?

We have added a more detailed description of the classes of autolysins in the introduction (lines 68-80).

Line 75: Please provide descriptions of “transenvelope nanomachine” as it is not a widely used term.

We have changed “transenvelope nanomachines” for “transenvelope complexes” and added an example, the type VI secretion system (line 91).

Line 81-82: I suggest rephrasing the heading since the paragraph primarily focuses on correlation between LD-crosslinking and LT activity based on measuring anhydromuropeptide released, but did not report the actual “glycan chain length” of PG.

We have changed the heading accordingly: “Peptidoglycan profile screening reveals a correlation between LD-crosslinking and anhydromuropeptide levels” (lines 116-117).

Line 90: To reference Supplemental Figure 2 together with Figure 1b since it displayed the entire heatmap.

We have added the reference to Suppl. Figure 2 (line 126).

Line 98-99: A lower concentration of CuSO₄ was used in the MM condition despite the working concentration being determined to be 1 mM (Supplemental figure 4c). Is 1mM of Cu²⁺ toxic to the cell in MM condition?

Yes, 1 mM CuSO₄ is toxic in MM. A preliminary assay to test the working concentrations showed 10 μM CuSO₄ was the maximum tolerated concentration of CuSO₄ (Fig. R8); hence, we have used 5 μM CuSO₄ in the experiments performed in MM, which effectively inhibits LD-transpeptidation.

Fig. R8: Analysis of CuSO₄ working concentrations in MM. OD₆₀₀ was measured in *V. cholerae* WT cultures grown in MM supplemented with increasing concentrations of CuSO₄ (0.0005 to 5 mM).

Line 111: It would be nice to indicate the p-values for Supplemental Figure 1b (LD crosslinking) and 1c to support the sentence “Further, despite comparable morphology and growth, ...”.

We believe the reviewer refers to Suppl. Fig. 6, since the figure referred to in the sentence in the original lines 111-113 was Suppl. Fig. 6bcd.

We cannot provide a p-value for LD-crosslink in Suppl. Fig. 6b as there are no crosslinks detected in the Δldt mutant sample and hence a statistical test cannot be performed.

We have included the p-values in Suppl. Fig. 6d.

All data in the plots are provided in the Source Data file.

Line 178: Please rephrase “Together, these results demonstrate that increasing the degree of LD-crosslinking can repel an attack by predatory LTs of both bacterial and phage origin.

Ok, we have rephrased the sentence (lines 228-230).

Line 203: "However, our results indicate that even low levels of LD-crosslinking can significantly inhibit LT activity". I think the authors are referring to Fig. 2c. It is interesting that Slt is inhibited more than other LTs. Perhaps I missed the point, but is it possible that it is due to Slt being an exo-LT?

We have added the reference to Fig, 2c for clarification (line 258).

We concur with the reviewer's perspective. Our research indicates that enzymes like Slt and MltB, which specialize in exolytic activity, are effectively regulated by lower levels of LD-crosslinking. On the other hand, enzymes such as MltD, with dual endo and exolytic functions, necessitate higher levels of LD-crosslinking to achieve similar inhibition effects. During our discussion, we emphasized the potential significance of the reported LD-crosslink accumulation at the chain ends as a key determinant in modulating the processivity of exolytic LT activities. It is also noteworthy to consider the possibility of other peptidoglycan structural elements playing a role in influencing LT activity. Moreover, it's important to acknowledge that while all assays were conducted under the same in vitro conditions, these may not necessarily represent the ideal conditions for each enzyme evaluated, suggesting an avenue for further optimization and exploration.

Line 213-215: "Our data indicate that PG containing DD-crosslinks is degraded more efficiently than PG containing LD-crosslinks." Please indicate the figure the authors is referring to.

We have added the reference to Fig. 2 (line 267).

Line 219-222: References that indicate *Agrobacterium tumefaciens*, *Mycobacterium tuberculosis*, or *P. damsela* PG having naturally high LD-crosslinked cell walls should be provided.

We have provided the references in line 277.

Line 242 and 254: The authors proposed that the LD crosslink could play a role in antibiotic resistance. However, in the high-throughput screen (Figure 1), the LD crosslink is not more pronounced under the antibiotic stress tested. Is there any reason for this observation?

We are not suggesting that LD crosslinks are directly implicated in antibiotic resistance. Rather, we propose that the modulation of LT activity by LD-crosslinks plays a pivotal role in the release of anhydromuropeptides, which is a known trigger for the upregulation of AmpC beta-lactamase expression in certain bacteria.

Our findings in *V. cholerae*, which the reviewer refers to, do not show an increase in LD-crosslinks due to antibiotic stress (Fig. S2). This aligns with our discussions on the adaptive nature of LD-crosslinking, which appears to be fine-tuned to specific ecological niches, rather than responding to a disrupted peptidoglycan synthesis-turnover balance, whether due to heightened LT activity (e.g., from phage attacks) or diminished synthesis (e.g., from beta-lactam interference).

Line 283: Indicate the abbreviation MCS. I believe it is "Multiple cloning site

The reviewer is correct. We have added the abbreviation in line 361.

Line 326: To correct the Parenthesis. Perhaps it is referring to "(100 µg/ml in water) overnight at 37°C."?

Yes, thanks for pointing out this typo. We have corrected it (line 406).

Line 533: To define what ND is in the figure legend. i.e. ND: Not detected.

We have added the meaning of ND in the figure legend (line 866).

Line 541-543: Figure 2d. How is the relative LT activity level calculated?

Thanks for pointing this out. To clarify, we have standardized the normalization of the LT activity across all figures for consistency. Initially, LT activity in Fig. 2d was benchmarked against the 100% activity observed on *V. cholerae* Δldt sacculi, which have 0% LD-crosslinks. This same reference point was applied to the in vitro assays for *P. damselae* WT and Δldt sacculi. In Fig. S9c, the baseline for 100% LT activity was set using *P. damselae* Δldt sacculi due to its higher activity levels.

To improve understanding and readability, we have recalibrated the LT activity in all relevant figures to be relative to the sacculi substrate with 0% LD-crosslinks. This adjustment ensures a uniform reference point across our data presentation. Consequently, we have updated Fig. 2cd, S9cd, and S10bc to reflect this change. We believe this provides a clearer and more coherent interpretation of our results.

Line 554-556: Figure 3d. What does '***' indicate?

We have added the meaning of ** to the figure legend (line 896).

Supplemental figure 10: Wrong indication of panel c in the figure legend.

Thanks, we have amended it.

Supplemental Figure 6c. Please correct the spacing issue. ' Δldt mutant cultures grown "overnight" in LB ...'

Thanks, we have corrected it.

Reviewer #4 (Remarks to the Author):

Reviewer #5 (Remarks to the Author):

Reviewer #6 (Remarks to the Author):

REVIEWERS' COMMENTS

Reviewer #1 (Remarks to the Author):

The manuscript by Cava and co- workers has been thoroughly revised. The authors have replied to all concerns raised by this reviewer.

There is only a minor comment to be addressed

Lane 306: "LdtAVc and LdtEEc controlled by RpoS and RpoE, respectively" should be "LdtAVc and LdtEEc controlled by RpoE and RpoS, respectively"

I have no further questions, and I recommend the manuscript for publication

Reviewer #2 (Remarks to the Author):

We appreciate the authors for addressing the comments constructively and performing additional experiments for the revised manuscript. However, the major concern still remains that the observations reported are due to the substrate-specificity of the various LTs, and may not represent any physiological regulation. This has been a major concern of this manuscript as pointed out by other reviewers as well. Although the authors have revised parts of the text to avoid terms implying regulation, a few important statements are particularly misleading, especially in the abstract/ summary. The experiments reported here do not support the general conclusions/ inference drawn.

Comments:

Line 35-36: "Moreover, we demonstrate that this regulation controls the release of immunogenic PG fragments and provides resistance against predatory LTs" – reiterating ours and other reviewers' point, regulation entails a stimulus and a response. While the Δ ldt mutant releases higher amount of anhydromuropeptides in the medium in comparison to a WT strain, whether Ldts are actively downregulated, and whether the released muropeptides are indeed immunogenic in the said environments, is yet to be studied. Similarly, the authors themselves have tested and not found evidence for increase in 3-3 crosslinks or Ldt expression upon phage infection. While this can be speculated in the discussion and is indeed an exciting possibility, it remains a speculation and should be clearly staged as such.

Line 97: "adaptive changes" should only be "changes", unless there is a disadvantage to the cells if these changes do not occur

Line 98: "widespread control mechanism", line 237: "widely conserved mechanism", line 249: "this regulatory mechanism": A mechanism would entail understanding how LT enzymes have a differential substrate specificity, with analysis of structure or enzyme kinetics. "Substrate specificity" is better suited here.

Line 179: “regulation of specific LTs” – activity of specific LTs

The inclusion of other phages in the susceptibility assay is a good addition. However, information regarding the basis for choosing these phages is relevant to the conclusions drawn, and should be provided in the text. In their rebuttal, the authors state “Initially, we examined the genomes of all phages within the collection, pinpointing those encoding endolysins. These were then categorized based on homology and functional domains. We finally selected a diverse array of phages, each representing a distinct endolysin type, and proceeded with our experiments.” This is an important analysis, and should be stated in the text. How clear is the distinction between lysozyme-like and LT-like enzymes in the in silico analysis? Are there prior reports on the nature of endolysins encoded by these phages which can be cited?

The experiment with overexpression of LdtE decreasing the phage lysis in *E. coli* needs few other controls. What happens in LdtE mutants? Is LdtE activity important for the protection? At what level the protection is happening? Again here, is this simply due to the decreased endolysin activity on 3-3 cross-linked peptidoglycan substrates?

The authors make a statement in their rebuttal letter regarding the significance of Ldt expression and phage susceptibility: “Our findings reveal that LD-crosslinking does not increase as a result of phage infection (Fig. R6B). These results are further supported by an RNAseq analysis of the transcriptional response of *E. coli* to lambda phage infection showed that none of the LDTs was differentially expressed (PMID: 37653008). Altogether, this suggests that while PG containing LD-crosslinks exhibits greater resilience to LT-mediated phage attacks, such resistance is not a direct response to the phage itself but is influenced by environmental factors, such as nutrient availability or outer membrane damage, as supported by previous research (PMID: 21792174, PMID: 30723128).” This important point has been left out of the results and not appropriately presented in discussion. It should be included in the results section.

Overall, I agree that authors show a very clear correlation between 3-3 crosslinks and LT activity. But where is the evidence of regulation and physiological significance? Unless, there is additional strong evidence shown for the regulation, the conclusions drawn in the manuscript can mislead the community.

Reviewer #3 (Remarks to the Author):

The authors have addressed all my concerns. I recommend publication.

Reviewer #4 (Remarks to the Author):

Reviewer #5 (Remarks to the Author):

Reviewer #6 (Remarks to the Author):

Response to remaining reviewers' comments

Reviewer #1 (Remarks to the Author):

The manuscript by Cava and co- workers has been thoroughly revised. The authors have replied to all concerns raised by this reviewer.

There is only a minor comment to be addressed

Lane 306: "LdtAVc and LdtEEc controlled by RpoS and RpoE, respectively" should be "LdtAVc and LdtEEc controlled by RpoE and RpoS, respectively

I have no further questions, and I recommend the manuscript for publication.

We thank the reviewer.

We have corrected the sentence (line 307).

Reviewer #2 (Remarks to the Author):

We appreciate the authors for addressing the comments constructively and performing additional experiments for the revised manuscript. However, the major concern still remains that the observations reported are due to the substrate-specificity of the various LTs, and may not represent any physiological regulation. This has been a major concern of this manuscript as pointed out by other reviewers as well. Although the authors have revised parts of the text to avoid terms implying regulation, a few important statements are particularly misleading, especially in the abstract/ summary. The experiments reported here do not support the general conclusions/ inference drawn.

Comments:

Line 35-36: "Moreover, we demonstrate that this regulation controls the release of immunogenic PG fragments and provides resistance against predatory LTs" – reiterating ours and other reviewers' point, regulation entails a stimulus and a response. While the Δ ldt mutant releases higher amount of anhydromuropeptides in the medium in comparison to a WT strain, whether Ldts are actively downregulated, and whether the released muropeptides are indeed immunogenic in the said environments, is yet to be studied. Similarly, the authors themselves have tested and not found evidence for increase in 3-3 crosslinks or Ldt expression upon phage infection. While this can be speculated in the discussion and is indeed an exciting possibility, it remains a speculation and should be clearly staged as such.

In this context, the term 'regulation' specifically refers to the inhibition of LT activity by LD-crosslinks, rather than addressing whether LDTs are regulated. Similar to many other species, *Vibrio cholerae* releases DAP-containing anhydromuropeptides, which have a well-established immunogenic effect (PMID 32677123). Our findings indicate that LD-crosslinks enhance cell wall resilience to exogenous LTs. While we did not observe any evidence that LDT expression changes in response to phage infection, numerous studies support the idea that these enzymes can be regulated under various environmental conditions. Although the environmental regulation of LDTs to bolster protection against predatory LTs falls beyond the scope of our study, we have briefly touched on this topic in the Discussion section (lines 304-318).

Line 97: "adaptive changes" should only be "changes", unless there is a disadvantage to the cells if these changes do not occur

We have removed "adaptive" (line 99).

Line 98: “widespread control mechanism”, line 237: “widely conserved mechanism”, line 249: “this regulatory mechanism”: A mechanism would entail understanding how LT enzymes have a differential substrate specificity, with analysis of structure or enzyme kinetics. “Substrate specificity” is better suited here.

We believe the term 'mechanism' is appropriately used in these sentences, as our data demonstrates that LD-crosslinks inhibit LT activity both in vitro and in vivo across multiple species in a dose-dependent manner. Although we have not conducted structural studies, our use of diverse PG-degrading enzymes (including various distinct LTs) offers valuable mechanistic insights into how LD-crosslinks broadly affect LT enzymatic activity, while leaving other PG-degrading enzymes, such as muramidases and endopeptidases, unaffected.

Line 179: “regulation of specific LTs” – activity of specific LTs

We have rephrased the sentence (line 180). Now it reads: “... indicating this crosslinking mode may be more effective at inhibiting specific LT enzymes in the cell.”

The inclusion of other phages in the susceptibility assay is a good addition. However, information regarding the basis for choosing these phages is relevant to the conclusions drawn, and should be provided in the text. In their rebuttal, the authors state “Initially, we examined the genomes of all phages within the collection, pinpointing those encoding endolysins. These were then categorized based on homology and functional domains. We finally selected a diverse array of phages, each representing a distinct endolysin type, and proceeded with our experiments.” This is an important analysis, and should be stated in the text. How clear is the distinction between lysozyme-like and LT-like enzymes in the in silico analysis? Are there prior reports on the nature of endolysins encoded by these phages which can be cited?

We have expanded the Methods section (lines 545-546) and included a supplementary table (Supplementary Table 4) with the phages genome and endolysin accession numbers and UniProt and CDD domain annotation.

The experiment with overexpression of LdtE decreasing the phage lysis in *E. coli* needs few other controls. What happens in LdtE mutants? Is LdtE activity important for the protection? At what level the protection is happening? Again here, is this simply due to the decreased endolysin activity on 3-3 cross-linked peptidoglycan substrates?

Overexpression of the well-characterized LdtE was utilized as a tool to increase LD-crosslinking levels in the host bacterium. Having confirmed the harmless effect of LdtE overexpression in *E. coli* (Suppl. Fig. 11), we chose to use *E. coli* with an empty plasmid as a control in comparison to LdtE overexpression, allowing us to evaluate the protective role of LD-crosslinks in phage assays. Additionally, we employed multiple phages encoding different PG-degrading enzymes to specifically test and verify the inhibitory effect of LD-crosslinking on LT enzymes.

The authors make a statement in their rebuttal letter regarding the significance of Ldt expression and phage susceptibility: “Our findings reveal that LD-crosslinking does not increase as a result of phage infection (Fig. R6B). These results are further supported by an RNAseq analysis of the transcriptional response of *E. coli* to lambda phage infection showed that none of the LDTs was differentially expressed (PMID: 37653008). Altogether, this suggests that while PG containing LD-crosslinks exhibits greater resilience to LT-mediated phage attacks, such resistance is not a direct response to the phage itself but is influenced by environmental factors, such as nutrient availability or outer membrane damage, as supported by previous research (PMID: 21792174, PMID: 30723128).” This important point has been left out of the results and not appropriately presented in discussion. It should be included in the results section.

While the environmental regulation of LDTs to bolster protection against predatory LTs falls beyond the scope of our study, we have briefly touched on this topic in the Discussion section (lines 304-318).

Overall, I agree that authors show a very clear correlation between 3-3 crosslinks and LT activity. But where is the evidence of regulation and physiological significance? Unless, there is additional strong evidence shown for the regulation, the conclusions drawn in the manuscript can mislead the community.

Reviewer #3 (Remarks to the Author):

The authors have addressed all my concerns. I recommend publication.

We thank the reviewer.

Reviewer #4 (Remarks to the Author):

Reviewer #5 (Remarks to the Author):

Reviewer #6 (Remarks to the Author):
